# Quiescent and Active Galactic Nuclei as Factories of Merging Compact Objects in the Era of Gravitational Wave Astronomy

**Manuel Arca Sedda** [1,2,3,*,†] ✪, **Smadar Naoz** [4,5,†] ✪ and **Bence Kocsis** [6,†] ✪

1    Dipartimento di Fisica e Astronomia "G. Galilei", Università di Padova, Via F. Marzolo 8, I-35131 Padova, Italy
2    Gran Sasso Science Institute (GSSI), I-67100 L'Aquila, Italy
3    Astronomisches Rechen-Institut, Zentrum für Astronomie der Universität Heidelberg, Mönchhofstr. 12-14, D-69120 Heidelberg, Germany
4    Department of Physics and Astronomy, University of California, Los Angeles, Los Angeles, CA 90095, USA
5    Department of Physics and Astronomy, Mani L. Bhaumik Institute for Theoretical Physics, University of California, Los Angeles, Los Angeles, CA 90095, USA
6    Rudolf Peierls Centre for Theoretical Physics, Clarendon Laboratory, Parks Road, Oxford OX1 3PU, UK
\*    Correspondence: m.arcasedda@gmail.com
†    These authors contributed equally to this work.

**Abstract:** Galactic nuclei harbouring a central supermassive black hole (SMBH), possibly surrounded by a dense nuclear cluster (NC), represent extreme environments that house a complex interplay of many physical processes that uniquely affect stellar formation, evolution, and dynamics. The discovery of gravitational waves (GWs) emitted by merging black holes (BHs) and neutron stars (NSs), funnelled a huge amount of work focused on understanding how compact object binaries (COBs) can pair up and merge together. Here, we review from a theoretical standpoint how different mechanisms concur with the formation, evolution, and merger of COBs around quiescent SMBHs and active galactic nuclei (AGNs), summarising the main predictions for current and future (GW) detections and outlining the possible features that can clearly mark a galactic nuclei origin.

**Keywords:** black hole physics; galactic nuclei; gravitational waves

## 1. Introduction

Most galactic nuclei (if not all) are expected to harbour in their centres either a dense nuclear cluster (NC), a massive stellar conglomerate typically comprised of $10^{4-9}$ stars [1–3], an supermassive black hole (SMBH), with typical masses $M_{\rm SMBH} \simeq 10^{4-10}$ M$_\odot$ [4,5], or both [6–10]. With densities up to several orders of magnitude larger than typical globular and young massive clusters, but similar half-mass radii, NCs constitute the densest stellar systems in the Universe [1,3,10]. The unique feature of hosting a massive NC or an SMBH—or possibly both—makes galactic nuclei the ideal laboratories to study stellar dynamics at its extreme. A particularly interesting aspect of dynamics in such extreme environments is the formation of compact object binaries (COBs), comprised of stellar-mass black holes (BHs) or neutron stars (NSs).

From the theoretical standpoint, the formation of COBs in galactic nuclei is rather uncertain, as it depends on many features, among which are the NC formation process, the SMBH influence, especially if it is in its active phase, the stellar evolution, and the secular dynamics. The high densities of galactic nuclei and NCs can favour stellar interactions, potentially boosting COB formation, but the large velocity dispersion associated with a central SMBH or a cuspy matter distribution hampers them, making it hard to assess the actual formation, and merger, rate of COBs in galactic nuclei. After formation, a COB can undergo further interactions with passing stars, which can lead to the binary shrinkage, to the replacement of one component, or to its disruption [11–13]. The SMBH gravitational field can alter the COB evolution, causing periodic oscillations of the binary

eccentricity, the so-called eccentric Lidov–Kozai (EKL) mechanism [14–21], which can ultimately lead to its coalescence. The gaseous disc surrounding SMBHs in active galactic nucleus (AGN) can strongly affect compact object (CO) dynamics, damping their orbits and boosting binary–single and single–single scatterings, possibly favouring binary formation (e.g., [22–35]).

From the observational point of view, the Milky Way's centre, with an SMBH weighing $M_{\mathrm{SMBH}} \simeq 4.1 \times 10^6$ M$_\odot$ [36–39] and an NC with a mass $M_{\mathrm{NC}} = 2.5 \times 10^7$ M$_\odot$ [12,40–42], constitutes an excellent, and our closest, observation site. Over a timespan of more than 30 yr, we have been capable of literally observing stars moving within a few thousand AU from the Galactic SMBH, SgrA*, with a precision sufficiently high to probe general relativity [36,43,44]. These observations permitted reconstructing the mass distribution in the inner 0.2 pc, suggesting that such a region would contain ~10,000 M$_\odot$ of BHs, and it is an unlikely nursery for an intermediate-mass black hole (IMBH) heavier than $10^3$ M$_\odot$ [39,45], supporting previous theoretical arguments (e.g., [46–52]). Further evidence of BHs at the Galactic Centre comes from recent observations of X-ray emission from a dozen X-ray binaries in the inner pc that suggest the presence of over 20,000 stellar BHs and NSs lurking there [53]. Despite this observational finding still being under debate [54], it finds support in several theoretical works [55–57] and seems a reasonable and natural outcome of stellar evolution for the Galactic NC [58,59].

Placing stringent constraints on the theoretical models of COB formation in galactic nuclei can enable us to create a diagnostic scheme to interpret their observations, especially with regard to stellar BHs in the era of gravitational wave (GW) astronomy.

In fact, despite the observations of BHs in binary systems becoming recently possible by observing the stellar companion's motion [60–62] and electromagnetic emission (EM) from accreted material [63,64], even in the Galactic Centre [53], GW detections represent so far the only known technique to provide uncontroversial proof of the existence of black hole binaries (BBHs). Since the ground-breaking discovery of GWs emitted by a merging BBH [65], the LIGO-Virgo-KAGRA collaboration (LVC) performed three observation runs and assembled a catalogue of almost 100 GW sources [65–78], making clear that the interpretation of these observations requires a profound rethinking of our understandings of COB formation and evolution. So far, the so-called GWTC-3 catalogue of GW sources includes mergers from ~60 BBHs, 2 double NSs, and 2 NS–BH binaries, the first merger with a total mass in the IMBH mass range above 100 M$_\odot$, and the first merger involving one object with a mass 2.6 M$_\odot$, either the lightest BH or the heaviest NS ever detected in a merging binary. With a number of detections that doubles the number of detections performed via EM emission and proper motion, the LVC BHs are becoming sufficiently numerous to enable placing constraints on the overall properties of the underlying population of BHs in merging binaries [77]. Interpreting the inferred properties of observed BBH mergers requires understanding the physics that regulates the formation and evolution of single and binary COs and their stellar progenitors, as well as the impact of different formation channels on their global properties.

Significant improvements in stellar evolution theories undoubtedly helped us to better understand the processes that govern the evolution of single and binary massive stars and how they can merge in galactic fields, despite the still many uncertainties about the CO mass spectrum, natal spins, and kicks (for an extensive review on the topic, see [79]). In particular, there are two aspects of stellar evolution particularly relevant to the formation and merger of COBs, namely the so-called *upper* and *lower* mass-gaps.

Generally, stars with zero-age main sequence (ZAMS) masses in the 22–26 M$_\odot$ range are expected to end their lives in a supernova (SN) event. If the SN explosion happens on a timescale of ~250 ms (rapid SN model [80]), the remnant will have a mass falling in the 3–5 M$_\odot$ range, whilst if the explosion timescale is the order of seconds, the star undergoes a failed SN and directly collapses to a BH with a mass above the 3–5 M$_\odot$ range (delayed SN model [80]). This opens a gap in the CO mass spectrum, called the lower mass-gap, whose existence is highly uncertain observationally (e.g., [81,82]) and intrinsically relies

on the uncertain physics of stellar evolution [79]. Heavier stars that develop a He core with a mass ∼64–135 $M_{\odot}$, instead, are expected to undergo an explosive process, the pair instability supernova (PISN), which rips the star apart and leaves no remnant [83,84]. Stars with a lighter core (32–64 $M_{\odot}$) develop rapid pulses that enhance mass-loss before the SN explosion—so-called pulsational pair instability (PPISN) [84,85]. These two processes, PISN and PPISN, cause a dearth of BHs with masses in the 40–150 $M_{\odot}$ range [85–90]. The extent of this "upper" mass-gap is rather uncertain, as it depends on stellar rotation, nuclear reaction rates, and accretion physics [91–95].

Broadly speaking, the formation channels of a merging COB are grouped into two main channels: isolated, i.e., a stellar binary paired at birth, which turns into a COB that eventually merges without the intervention of other objects, and dynamical, i.e., a COB assembled dynamically, which eventually merges with the aid of multiple gravitational scatterings in star clusters. Unfortunately, the localisation accuracy of current GW detectors is too low to pinpoint the location of the merger event, thus generally, the impact of different formation channels on the overall merger population is assessed on a statistical basis. Several recent works tried to untangle the signatures of different formation scenarios in BBH merger populations by looking at different parameters that can be retrieved from GW detections—such as the component mass, chirp mass, effective spin parameter—but this hugely depends on many uncertainties: the cosmic star formation history or metallicity distribution, BH natal spins and mass spectrum, the physics of single and binary stellar evolution, the properties of star clusters at birth, and the physics of galactic nuclei [96–102].

The mass of the merging objects represents one of the possible quantities that can be used to discern between an isolated and dynamical origin. BBHs formed in isolation are expected to have components with a mass below 40–60 $M_{\odot}$, owing to the PISN and PPISN mechanisms (e.g., [86,103]), thus making it hard to explain the existence of upper mass-gap objects—such as the one observed in the GW190521 source [76]—via isolated stellar evolution only. Upper mass-gap objects are easier to form in dynamically active environments, such as star clusters and galactic nuclei, where they can form either via stellar mergers [104–110], BH–star accretion events [105–107], or repeated—so-called *hierarchical*—mergers [98,99,105,111–116]. These processes are affected by uncertainties though: the stellar evolution of post-merger stars is poorly known and requires detailed hydrodynamical simulations (e.g., [109,110,117]); the fraction of stellar matter actually accreted onto a BH in a star–BH collision is rather unknown [118–122]; repeated mergers are hampered by post-merger GW recoil kicks [123–125], which can eject the remnant BH from the host cluster and avoid further mergers [98,98,126–129]. Similarly, the development of mergers with one component in the lower mass-gap, such as GW190814 [74], seems to be unlikely in isolated binary models [100], whilst they can more easily form dynamically [130–136].

The level of spin alignment represents another quantity useful to discern isolated and dynamical mergers. In fact, isolated mergers are expected to feature (approximately) aligned spins [137–139], whilst in dynamical mergers, the chaotic process forming the binary is expected to distribute the spin–orbit angle isotropically [99,102,138,140].

Hierarchical mergers, which are a natural by-product of the dynamics in high-density environments, might have unique mass–spin features; thus, a combined statistical analysis of such quantities could help in determining whether an observed GW source originated from the remnants of previous merger episodes. For example, assuming that the cosmic population of merging BHs is characterised by a spin distribution peaked at relatively low values, as indicated by GW observations [141], implies that a population of second-generation mergers will be inevitably characterised by larger masses and higher spins, thus being clearly discernible from the population of first-generation mergers and possibly from isolated mergers as well [99,142].

Aside from masses and spins, there is a further binary parameter that could represent a smoking gun of a dynamical origin, the binary orbital eccentricity. Placing constraints on the eccentricity of observed mergers became possible only recently (e.g., [143–146]) and led to placing constraints on the eccentricity of up to four LVC sources, for which

possibly $e > 0.1$ [147–150]. Interestingly, for eccentric sources, the accuracy of the parameter measurement can significantly increase, e.g., the chirp mass (localisation) accuracy of an eccentric 30–30 $M_\odot$ BBH can be $\sim 10^1 (10^2)$-times higher than for the circular case [151]. Generally, LVC detectors at design sensitivity could distinguish between circular and eccentric models provided that $e_{10\,Hz} > 0.04$, the measurement error on the eccentricity being around $\delta e \sim (10^{-4} - 10^{-3})(D/100\mathrm{Mpc})$ [151,152].

In isolated COBs, several processes (e.g., tidal interactions or dynamical friction during a common envelope phase) tend to circularise the orbit of the binary progenitor (e.g., [153–157]), although some of the physical processes that are still partly unknown—such as the common envelope—could produce mildly eccentric binaries [158]. Conversely, the eccentricity distribution of dynamically assembled BBHs generally follows a thermal distribution, $P(e)de \sim 2e$, which implies a probability of 50% to form a binary with eccentricity $e > 0.7$. Theoretical models predict that around 1–10% of mergers forming in globular clusters can have an eccentricity $e > 0.1$ in the frequency band typical of LVC and ground-based GW detectors (i.e., 10 Hz) [138,159–161], and up to 30% could be eccentric in the LISA band ($\sim 10^{-3}$ Hz) [162–164].

What about galactic nuclei? In such complex environments, the formation of COBs is regulated by a variety of dynamical processes—some acting in concert, others acting in contrast—which intrinsically affect the overall properties of those that eventually merge and may leave imprints that could differ from the general expectations of the dynamical channels.

This review aimed at providing a broad overview of the processes that can aid or hamper COB formation and mergers in galactic nuclei harbouring a central SMBH, either in its quiescent or active phase, possibly surrounded by an NC, and discussing the main properties and detection prospects of GW sources formed in galactic nuclei. We organised the review according to the main phases characterising the evolution of a galactic nucleus, following a zoom-in scheme, from the possible formation of the central NC to the coalescence of COBs:

- We start by briefly reviewing the current knowledge on the observational evidence of single and binary COs in galactic nuclei (Section 2);
- We describe how the environment, stellar evolution, and dynamics can affect CO populations (Sections 3 and 4);
- Moving closer to the SMBH, we describe the main dynamical processes at play to form COBs in galactic nuclei (Section 5) and discuss the impact of secular processes in quiescent galactic nuclei (Section 6) and gaseous effects in AGNs (Section 7) on the formation of merging COB;
- Finally, we discuss the main properties of merging COBs in galactic nuclei, focusing on the prospects for current and future GW detections (Section 8).

Figure 1 sketches the main phases of COB formation in galactic nuclei and provides a schematic and ordered illustration of the themes touched on in this review.

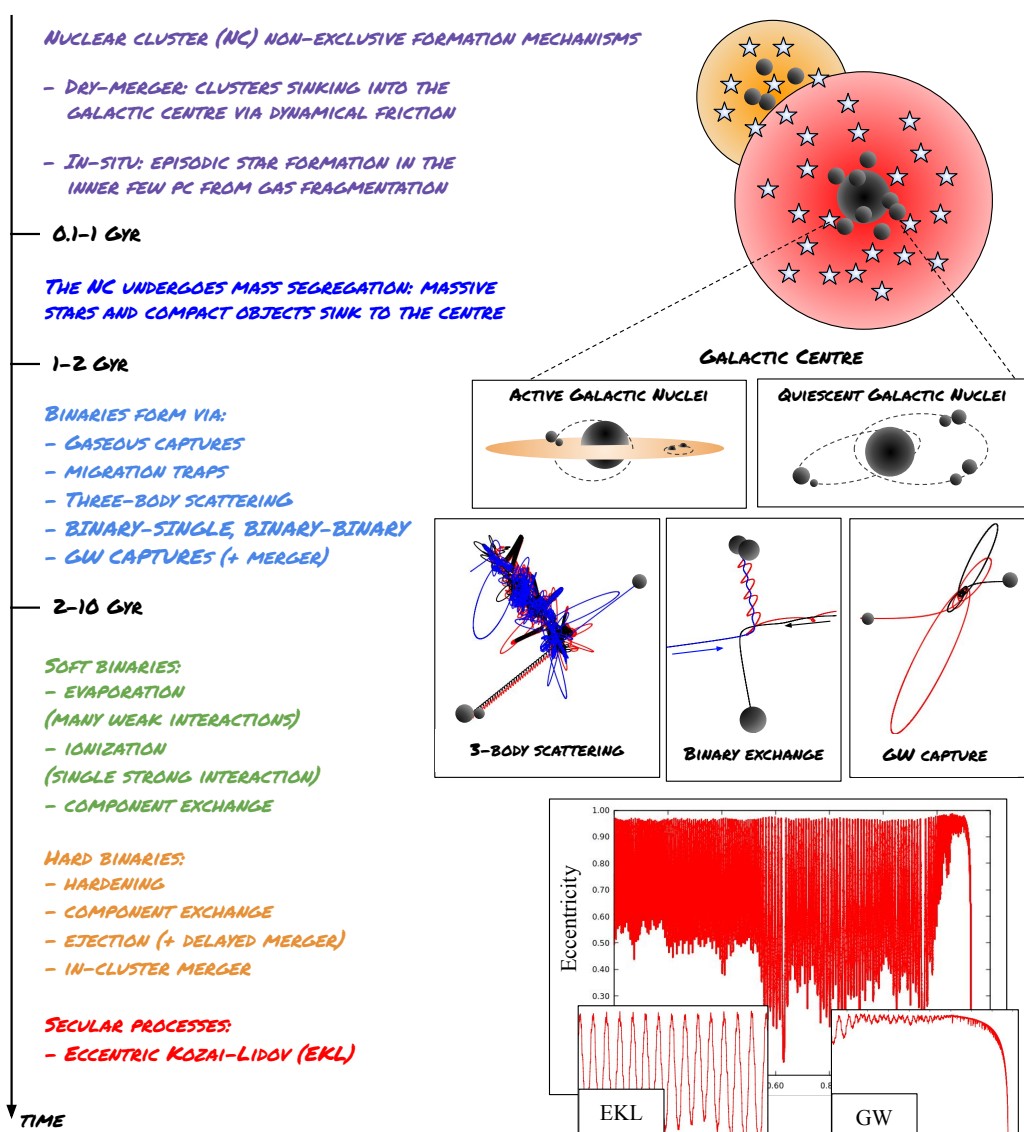

**Figure 1.** Schematic illustration of the galactic nuclei zooniverse: a nuclear cluster (NC) forms via in situ star formation and mass transport from infalling star clusters; in its inner parts, a variety of dynamical interactions can trigger the formation of compact object binaries (COBs); in other cases, COBs promptly merge, releasing gravitational waves (GWs); in some other cases, the COB evolution is determined by the central supermassive black hole (SMBH) gravitational field, which can impinge periodic oscillations on the binary eccentricity and ultimately lead to their coalescence. The depicted interactions are actual *N*-body simulations carried out with the `ARGdf` code described in Arca-Sedda and Capuzzo-Dolcetta [165].

## 2. Observational Evidence of Binaries in Galactic Nuclei: The Milky Way Test Case

Most, if not all, high-mass stars *in the field* reside in a binary or higher multiple configurations [166–169]. Although challenging, observations of the inner pc of the Milky Way revealed the presence of a handful of spectroscopic and eclipsing binaries comprised of massive OB and Wolf–Rayet (WR) stars [170–174]. These observations suggest that the fraction of spectroscopic binaries attains values $f_b \simeq 0.34$, whilst this quantity drops to $f_b \sim$0.03–0.04 for eclipsing binaries, similar to the binary fraction inferred in young clusters. More in general, the aforementioned observations suggest that the total fraction of massive binaries in the Galactic NC is comparable to the field (e.g., [170,172]). The observation of X-ray sources [53,175] and diffuse X- (e.g., [176–178]) and $\gamma$-ray [179–186] emission at the Galactic Centre support the presence of COBs in galactic nuclei, although it

is unclear whether they are related to X-ray binaries, millisecond pulsars, or cataclysmic variables (e.g., [53,56,59,175–178,187,188]).

Complementary theoretical and modelling studies of these observations also suggested that the binary fraction in the inner 1 pc at the centre of the Galaxy is high. For example, star formation models suggest that the in situ formation of stars and, thus, binaries is expected at the centre of galaxies (e.g., [189,190]). Hyper-velocity stars (e.g., [191–195]) may imply the existence of binaries that arrive on a trajectory nearly radial to the tidal breakup radius of the SMBH, known as the Hills mechanism [196]. This mechanism may eject one star at a high velocity, while the other one may be captured on an eccentric orbit close to the SMBH, which was proposed to explain the existence of the S-cluster (e.g., [58,197–202]).

Other theoretical and observational analyses suggest that binaries can remain stable for a long time even when interacting with neighbouring stars (e.g., [173,203]). Thus, over the age of the young stars of the nuclear star cluster, estimated as a few Mys [204], about 70% of the binaries may retain their binary configuration [205]. Lastly, it was suggested that some of the peculiar features of the stellar disc in the Galactic Centre [206,207] can be explained by the possible observed high fraction of binaries [208].

### 3. Environmental Effects on Binary Formation in Galactic Nuclei

How can COBs form in galactic nuclei? This is one of the key questions that we tried to address in this review. This section is devoted to discussing how the presence of pristine stellar binaries and the formation process of the galactic nucleus can impact the formation of COB and COBs.

### 3.1. Binaries in Galactic Nuclei: Primordial, Dynamical, or Hybrid?

Generally, it is possible to distinguish two main formation channels for stellar and CO binaries: either the binary components were already paired at birth, in which case the binaries are called *primordial*, or they found each other via multiple interactions with other stars and COs forming *dynamical* binaries. In dense stellar environments, such as massive clusters and NCs, a further possibility suggests that primordial binaries underwent interactions with other members of the galactic nucleus and either suffered orbital modifications or exchanged one of their components; thus, they constitute a *hybrid* class, halfway between purely primordial and dynamical binaries.

Either way, after their formation, the evolution of these binaries will inevitably be affected by dynamical encounters, which in galactic nuclei are typically more frequent and violent than in galactic fields.

During each subsequent interaction, binaries and single objects suffer a change of their energy and angular momentum, up to a point where the system will have lost memory of its initial conditions. This process, called relaxation, occurs on a timescale roughly given by [209,210]

$$t_{\text{relx}} = 4.2 \, \text{Gyr} \left( \frac{15}{\log \Lambda} \right) \left( \frac{R_h}{4} \right)^{3/2} \sqrt{\frac{M_c}{10^7 \, \text{M}_\odot}}, \tag{1}$$

where $\log \Lambda$ is the Coulomb logarithm, $R_h$ is the cluster half-mass radius, and $M_c$ is its total mass.

There are a plethora of dynamical processes that can concur with the formation and evolution of binaries in a dense galactic nucleus. Some of them involve single or multiple dynamical interactions with other stellar objects (see Section 5), such as GW bremsstrahlung in single–single encounters (cap), three-body encounters (3bb), binary–single (bs) and binary–binary (bb) encounters, and triple evolution (3ev). Others involve secular effects impinged by the galactic nucleus morphology, the gravitational field of the central SMBH, or the collective effect of all stars orbiting around the SMBH (see Section 6), including the eccentric Kozai–Lidov (EKL) oscillations, general relativistic effects arising from the motion around the SMBH (1pN), and scalar (rr,s) and vector resonant relaxation (rr,v) processes,

which torque the orbit of the $i$-th star owing to the overall perturbation from all the $N - i$ stars in the nucleus.

The concurring action of one process or another is regulated by their typical timescales, which can vary over many orders of magnitude depending on the galactic nucleus's properties. Figure 2 briefly summarises how the timescales connected to these different mechanism vary at varying COB separation and assuming a COB with mass $M_{COB} = 20 \, M_\odot$ and an SMBH with mass $M_{SMBH} = 10^6 \, M_\odot$. To make the plot more readable, we do not include the typical AGN lifetime, which is expected to be a constant value in the range 1–100 Myr, and the three-body scattering time, which, for this specific example, is several orders of magnitude larger than all other timescales. From the plot, we see that dynamical friction and some of the secular effects, such as EKL, clearly operate over timescales shorter than the typical timescale of stellar evolution for CO progenitors (i.e., $\sim 1$–10 Myr), suggesting that dynamics may play a role even before massive stars turn into COs.

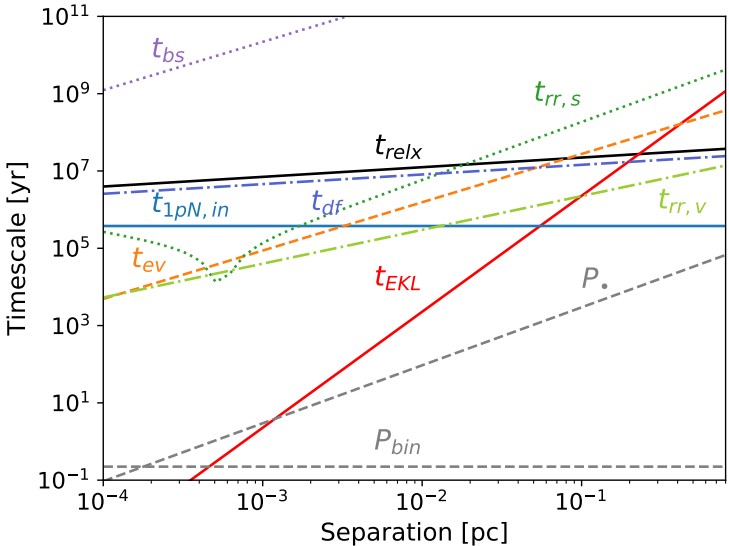

**Figure 2.** Timescales of several mechanisms affecting the formation of single and binary compact objects in a galactic nucleus similar to the Milky Way. Here, we assumed a binary mass $M_{COB} = 20 \, M_\odot$ and a supermassive black hole mass $M_{SMBH} = 10^6 \, M_\odot$. Different lines identify different processes: 2-body relaxation ($t_{relx}$), scalar ($t_{rr,s}$) and vector ($t_{rr,v}$) resonant relaxation, evaporation ($t_{ev}$), dynamical friction ($t_{df}$), binary–single scattering ($t_{bs}$), eccentric Kozai–Lidov ($t_{EKL}$), general relativistic precession ($t_{1pN,in}$), binary period ($P_{bin}$), orbital period of the binary about the supermassive black hole ($P_\bullet$). For the sake of visibility, the plot does not include the AGN lifetime (nearly constant line, 1–100 Myr) and the three-body and gravitational wave capture (single–single) scattering time, which are much larger than all other timescales for the adopted example.

This dynamical process has crucial implications for the evolution of primordial binaries in massive galactic nuclei like the one in the Milky Way. For example, it has been shown that a population of stellar binaries, in a Milky Way-like NC, would be strongly affected by both dynamical and secular mechanisms, undergoing several possible pathways that take place prior to CO formation, such as (see, e.g., [59,205]): (a) dissociation (75% of the population), (b) merger induced by eccentric Kozai–Lidov oscillations (10%), (c) shrinkage and circularisation of the orbit due to tidal synchronisation (13%), or (d) radial mergers and nearly head-on collisions (2%).

Among binaries that shrunk via stellar evolution—i.e., Case (c) above—only a fraction of 0.5% decouple from the SMBH dynamics and evolve into COBs that possibly merge within a Hubble time [59]. Note that this is assuming that the Galactic NC contains $\sim 2.5 \times 10^7$ stars and a primordial binary fraction $f_b = 0.3$ implies a number of *primordial* merging COBs of around $N_{COB} \sim 2500$, although such an estimate certainly depends

on the details of binary stellar evolution. The population of merging COBs formed from binary stellar evolution is expected to be comprised of BBHs (∼75%) and NS–BH binaries (∼25%), whilst double-NSs are unlikely to form [59]. These binaries may further interact with neighbouring single objects, possibly leading to the formation of new binaries via component exchange or the binary abundance may be reduced following ionization by strong encounters.

*3.2. Nuclear Cluster Formation Processes: In Situ Versus Dry Merger*

In Section 3.1, we briefly described how primordial binary formation and dynamical and secular processes can affect the formation of stellar and CO binaries. The dominance of one process over another intrinsically depends on the environment. For example, the density and velocity dispersion in the nucleus strongly affect the interaction rate of few-body interactions, whilst the matter density distribution determines how likely it is for a star to orbit within a given distance from the central SMBH, thus affecting the development of secular effects such as EKL indirectly.

In these regards, the presence of an NC in the galactic centre, which is clearly much denser than the stellar distribution of the galactic field, can substantially affect the formation and evolution of both single and binary COs. As we will see in this and the following sections, high densities and large masses make NCs potential factories of COs and COBs and might provide a significant contribution to the overall population of GW sources, but assessing the actual properties of "nuclear" COB mergers demands understanding how the macro-physics that regulates NC formation affects the micro-physics that governs the COs and COBs in galactic nuclei.

NCs are pretty peculiar objects, which exhibit a clear flattened morphology and rotation [3,8,211–215], a complex star formation history and multiple stellar populations [216,217], and in some cases, harbour a central SMBH [6–8,218].

Although the actual fraction of NCs containing an SMBH is rather uncertain [3], this sub-class of NCs represents, as we will discuss in the next section, the ideal place to study a huge variety of dynamical processes acting simultaneously. Further insights into NCs harbouring an SMBH come from a special class of objects called ultracompact dwarf galaxies (UCDs) [219]. Generally, a UCD is a compact stellar system characterised by a complex star formation history, a central SMBH, and an associated stellar stream [220]. These objects are thought to be the relics of galactic nuclei stripped in a galaxy merger event [221,222], and their properties apparently overlap with those of massive globular clusters and NCs [223]. If UCDs are what remain of NCs stripped away from their parent galaxies, the fraction of NCs harbouring an SMBH could be as large as 75–90% [224]. Generally, the mass of NCs and SMBHs seems to linearly scale with the host galaxy stellar mass [1,225,226]. Moreover, the SMBH-to-NC mass ratio increases with the galaxy mass [3], and NCs become too faint to be identified in galaxies with masses $M_g \gtrsim 10^{10}$ M$_\odot$, i.e., when the SMBH-to-NCs mass ratio becomes larger than 1 [7].

These interesting peculiarities may be intrinsically related to the processes that regulate NC formation, which are still partly unknown.

There are currently two most-debated formation scenarios for NCs: in situ and dry merger. In the in situ scenario, gas funnelled toward the galactic centre, e.g., owing to the effect of the galactic bar or inhomogeneities associated with a galaxy merger, cools and fragments, forming stars in a dense NC [227–236]. The in situ scenario may explain some of the features observed in NCs, such as rotation and flattening, the development of multiple episodes of star formation, and the absence of NCs in galaxies heavier than $10^{11}$ M$_\odot$, where the radiative feedback of the central SMBH becomes sufficiently strong to quench star formation and hamper NC growth (e.g., [233]).

The basic idea behind the dry merger formation scenario, instead, is that massive bodies travelling through a sea of lighter particles, such as a star cluster moving in the galaxy field, experience a drag, called dynamical friction, that slowly forces it to spiral inward and ultimately collide to form an NC [223,235,237–244]. The typical timescale of

this process can be relatively short (∼0.1–1 Gyr) [242,243,245]. As the clusters move inward, the increasing tidal force exerted from the galactic field and possibly from the central SMBH strips their outer regions, while their core keeps spiralling in. The competing action between tidal forces and dynamical friction ultimately determine whether spiralling clusters can efficiently build up an NC. Observations [7,37], simulations [246], and semi-analytic models [235,242,245] seem to support this picture, suggesting a typical host galaxy virial mass, $M_g \gtrsim 10^{11}$ M$_\odot$, above (below) which nuclei are dominated by a central SMBH (NC). Therefore, both the in situ and dry merger scenario may be able to explain (some) observational features of NCs, making it difficult to identify a single dominant process. In fact, the most-recent works indicate that the in situ and the dry merger scenarios work in concert, with the former possibly leading to NC formation in galaxies heavier than $10^9$–$10^{10}$ M$_\odot$ and the latter being the dominant channel in smaller galaxies [215,235,246–249].

Interestingly, it seems that the average age of NC stars increases at decreasing host galaxy masses [215,248,249], indicating that NCs in galaxies like the Milky Way likely formed through either a recent star formation burst (in situ) or the collisions of young massive clusters born close to the galactic nucleus (dry merger), whilst NCs in dwarf galaxies might be the relics of ancient cluster collisions or star formation occurring during an early stage of the galaxy's life.

Do NCs form first and SMBHs grow later or vice versa? This basic "chicken or egg" question is still unanswered. From the observational point of view, while SMBH candidates have been observed up to redshift $z \gtrsim 7$ [250], it is practically impossible to distinguish an NC at those cosmological distances, thus making it impossible to date the oldest and farthest NCs. From the theoretical point of view, some works propose that *primordial* NCs in high-redshift galaxies can sustain the formation of SMBH seeds via BH merger and accretion events (e.g., [251–254]), whose subsequent growth could evaporate the NC. Whilst this possibility may explain the dearth of NCs in the most-massive galaxies, it is at odds with their observation in low-redshift ($z < 1$) galaxies with masses $M_g < 10^{11}$ M$_\odot$, thus suggesting that there may be different NCs formation processes. Other theoretical works, instead, propose that the SMBH forms first and an NC forms later, either via the fragmentation of gaseous clouds [233] or star cluster infall [238]. For simplicity, in the following, we assumed that SMBHs are already fully grown at NC formation.

The Impact of Nuclear Cluster Formation Scenarios on the Population of Compact Objects in Galactic Nuclei

How can the NC formation scenario affect the formation of single and binary COs?

In the case of the in situ scenario, stellar and CO binaries form either primordially during the star formation process or dynamically via multiple encounters. The total number of CO progenitors is set by the NC mass and the adopted initial mass function (IMF) of stars and could be possibly enriched by multiple star formation episodes over cosmic history. In this sense, both COs and their progenitors form in the same environment. Let us consider an NC formed purely through the in-situ mechanism, with a total mass of $M_{NC} = 10^7$ M$_\odot$ and an average stellar mass of $m_* = 1$ M$_\odot$. Assuming a typical initial mass function [255] and considering the fact that COs form from the death of stars heavier than $\gtrsim 18$ M$_\odot$, we expect that COs constitute a fraction $f_{BH} \sim 10^{-3}$ of the whole population. Therefore, the total number of BHs harboured in our in situ NC should be simply given by

$$N_{BH,in} = f_{BH} M_{NC} / \langle m_* \rangle = 10,000 . \tag{2}$$

The population of NSs, instead, will be significantly affected by two processes. The first is related to the fact that NSs at birth receive a natal kick, which in 60–90% of the cases exceeds 100 km s$^{-1}$; thus, NSs would be promptly ejected even if they form in a dense NC. Note that the problem of NS natal kicks is still actively debate. Observations of Galactic pulsars suggest that the distribution of NS kicks is either Maxwellian, with a dispersion of $\sigma = 265$ km s$^{-1}$ [256], or bimodal [257–259]. From the theoretical standpoint, recent models focused on SN explosion predicts that the kick amplitude depends on the mechanism

that trigger the explosion (e.g., [260,261]), with the so-called electron-capture SN (ECSNe) possibly being the main source of NS retention in star clusters [262]. Moreover, once BHs settle in the galactic centre owing to mass segregation, they will prevent the segregation of lighter stellar species, pushing them onto wider orbits (e.g., [133,263]). Since the fraction of stars turning into an NS is expected to be roughly $f_{NS} = 10^{-3}$, we would expect only $N_{NC,in} = f_{ret}f_{NS}M_{NC}/\langle m_* \rangle = 6000$–9000, with $f_{ret} = 0.6$–0.9 the NS retention fraction for an escape velocity of 100 km s$^{-1}$. Despite being much lighter than BHs, also NSs may follow a cuspy surface density profile with $r^{-1.5}$, as shown in [264–266].

Let us consider now the dry merger scenario.

In the case in which the galactic nucleus is entirely made up of spiralled star clusters, the total amount of mass in BHs brought there will represent a fraction $f_{BH}$ of the cluster mass. If the BHs were mass segregated in the parent cluster, it is reasonable to assume that $M_{BH} \sim f_{BH}M_c$. However, owing to the galactic field, only a fraction $f_i$ of the clusters' mass will reach the centre and build up the NC, $M_{NC} = \sum_i f_i M_c$.

Let us assume that a population of $N_c = 20$ clusters with mass $M_c = 10^6$ M$_\odot$ fall in a galactic nucleus and lose a fraction $f_i = 0.5$ of their initial mass (see, e.g., [48]). The NC mass will thus be $M_{NC} = f_i N_c M_c$. In the process, the clusters will bring into the growing nucleus a fraction $f_{BH} \sim 10^{-3}$ of their total number of stars in the form of BHs, because BHs are likely segregated in the cluster core, and they will not be affected by the cluster mass-loss process. Thus, the number of BHs lurking in the final nucleus will be equal to

$$N_{BH,dry} = f_{BH}N_{cl}M_{cl}/\langle m_* \rangle = 20,000, \tag{3}$$

around twice compared to the mass inferred for an in situ NC, although the difference in the numbers of BHs can be substantially affected by a number of unknown quantities, such as the star cluster mass function and galactocentric distribution, the SMBH mass, or the amount of cluster mass actually brought into the galactic centre.

In the case of the dry merger scenario, the timescale over which clusters collide to form an NC can crucially determine whether the dynamics and stellar evolution processes have already affected the population of COs in the infalling clusters. This aspect is particularly important for the population of NSs that can be transported into a galactic nucleus through this process. For clusters with escape velocity $\sim$40 km s$^{-1}$, it has been shown that only 5–10% of NSs receives a kick sufficiently small to be retained [262,267]. If the infall process proceeds slower than stellar evolution, NSs will form in their parent cluster, and most of them will likely be ejected well before reaching the galactic nucleus. If we assume that only a fraction $f_{ret} \sim 0.05$–0.1, the number of NSs that can be accumulated in a Milky Way-like nucleus is $N_{NS,dry} \sim f_{ret}f_{NS}N_{cl}M_{cl}/\langle m_* \rangle \sim 1000$–2000, thus a factor 4–5 smaller than in the case of the in situ NC formation scenario. Suppose the infall process, instead, is faster than the stellar evolution process. In that case, stars will evolve into NSs after settling into the galactic centre, and their retention fraction will likely be similar to the one inferred for the in situ process, in which case, $N_{NS,dry} \sim 12,000$–18,000.

Future observations capable of providing insights into the population of BHs and NSs at the Galactic Centre could thus provide crucial information about the NC formation history [268], as different formation channels are expected to produce a substantially different population of COs.

Clearly, the arguments above serve as an order of magnitude estimate, and a more detailed approach is needed to fully characterise the properties of COs in galactic nuclei and the processes operating there.

Given a star cluster with mass $M_c$, orbital radius $r_c$, and eccentricity $e_c$, dynamical friction will drag it into the galactic centre over a timescale [243,269]

$$t_{df} = 0.3\text{Myr}\left(\frac{R_g}{1\text{kpc}}\right)^{3/2}\left(\frac{M_g}{10^{11}\ \text{M}_\odot}\right)^{-1/2}\left(\frac{M_g}{M_c}\right)^{0.67}\left(\frac{r_c}{R_g}\right)^{1.76}f(e_c,\gamma), \tag{4}$$

where $M_g$, $R_g$, and $\gamma$ represent the galaxy total mass, length scale, and slope of the density profile. The term $f(e_c, \gamma)$ is a function of the infaller orbital eccentricity and the density slope [243]:

$$f(e_c, \gamma) = (2 - \gamma)\left[a_1\left(\frac{1}{(2-\gamma)^{a_2}} + a_3\right)(1 - e_c) + e_c\right],$$ (5)

where $a_1 = 2.63 \pm 0.17$, $a_2 = 2.26 \pm 0.08$, and $a_3 = 0.9 \pm 0.1$. It is worth noting that this simple expression for the dynamical friction timescale represents a relatively good approximation also for COs orbiting inside a massive star cluster [263]. As the cluster orbits around the galactic centre, and slowly sinks, its internal dynamics will be regulated by several internal processes, the earliest of which will be mass segregation [209,270], a mechanism driven by dynamical friction [271] by which the most massive stars rapidly segregate toward the cluster centre [270]. The mass segregation timescale can be expressed as [210]

$$t_{\text{seg}} = 0.42\,\text{Gyr}\left(\frac{10m_*}{m_{\text{CO}}}\right)\frac{t_{\text{relx}}}{4.2\,\text{Gyr}}\left(\frac{r_{\text{CO}}}{R_h}\right)^{3/2},$$ (6)

where $m_{*,\text{CO}}$ is the average mass of stars (COs) and $t_{\text{relx}}$ is the half-mass relaxation time. Mass segregation gathers the most-massive objects into the inner cluster regions, favouring the development of strong interactions and the ejection of the most-massive components. This mechanism is particularly effective once BHs have formed and settled in the centre of the cluster. In fact, owing to their cross-section, larger than for a "normal star", the most-massive BHs tend to undergo the strongest interactions in cluster nuclei, pairing together and ejecting each other from the parent cluster in what is called the *BH burning process* (e.g., [272,273]). As a consequence, the internal dynamics will have time to affect the CO population in star clusters with sufficiently short segregation times, i.e., $t_{\text{seg}} < t_{\text{df}}$, before they reach the galactic centre and collide to build up the NC. If the segregation time is even shorter than the timescale of stellar evolution for massive stars, though, interactions among the most-massive stars can trigger runaway stellar collisions and possibly the formation of a very massive star that ultimately can collapse into an IMBH (e.g., [274–276]).

Figure 3 compares the dynamical friction and mass segregation timescales for a population of COs with mass $m_{\text{CO}} = 5$–50 $M_\odot$ inhabiting star clusters with a mass in the range $M_c = 10^{4-7}\,M_\odot$ and half-mass radius $R_h \sim 1\text{pc}(M_c/M_\odot)^{0.13}$ [277], orbiting at a distance of $r_c = 50 - 100$ pc in a galaxy with total mass $M_g = 10^{11}\,M_\odot$, scale radius $R_g = 1$ kpc, and the slope of the density profile $\gamma = 0.5$.

The plot suggests that the population of massive stellar objects has already been "dynamically processed" in clusters with a mass $M_c < 10^5\,M_\odot$ when they reach the galactic centre, whilst the population in the heavier cluster should be more representative of the cluster's initial population. Let us consider a population of $N_{\text{dec}}$ clusters each composed on average of $N_* = 10^6$ members, a fraction $f_{\text{BH}}$ of which are in COs and a fraction $\eta$ of their COs in binaries. If a fraction $\delta$ of all COBs survives the cluster infall, the total number of delivered COBs via the dry merger mechanism will be [48]

$$N_{\text{dec}} = 2000\delta\left(\frac{N_{\text{dec}}}{20}\right)\left(\frac{f_{\text{BH}}}{10^{-3}}\right)\left(\frac{N_*}{10^6}\right)\left(\frac{\eta}{0.1}\right).$$ (7)

Note that, if the infalling clusters are "dynamically young", meaning that the cluster relaxation time is much shorter than the infall time, their CO population will still be unaffected by dynamical processes and a fraction $\delta = 0.7 - 0.88$ of their BH population can be brought to the galactic centre [48].

Once COs are formed, or are brought, in the galactic centre, their subsequent evolution will mostly be driven by the dynamics, making it hard to distinguish objects formed in situ or delivered by infalling clusters. Possible differences may arise in the number and orbital properties of COBs and the mass spectrum of COs. For example, dynamical processes will have had time to substantially affect the population of BHs and NSs in clusters falling into the NC over timescales longer than the clusters' dynamical times, likely

reducing the number of COs and the average BH mass—owing to the BH burning process. Exploring how different CO populations in in situ or dry merger NCs are is difficult, owing to the underlying uncertainties, and in fact, generally in the literature, the initial properties of single and binary COs relies upon agnostic guesses. Devising and developing self-consistent NC models that implement both the NC formation process and the detailed stellar dynamics and evolution is key to shedding light on the fingerprints of NC formation history on the population of single and binary COs in galactic nuclei.

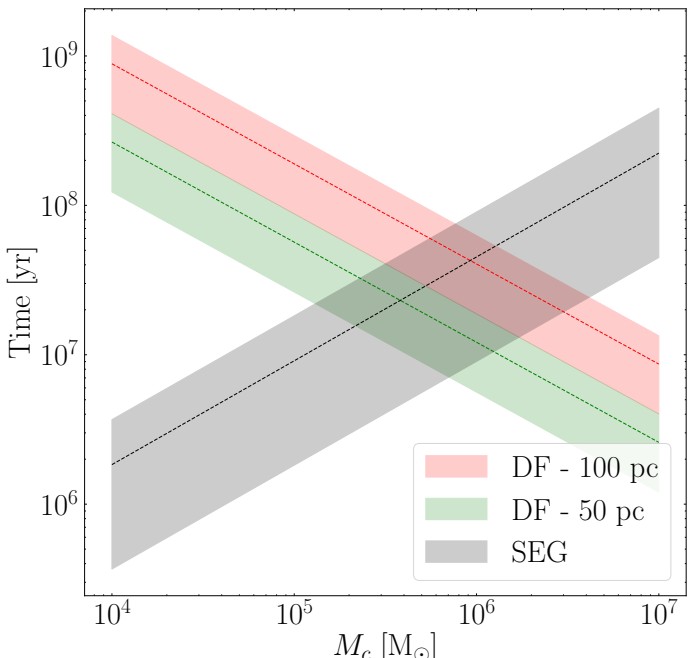

**Figure 3.** Dynamical friction timescale as a function of the cluster mass assuming that the cluster orbits at 50 pc (green) or 100 pc (red) in a galaxy with total mass $M_g = 10^{11}$ M$_\odot$, scale radius $R_g = 1$ kpc, and the slope of the density profile $\gamma = 0.5$. For comparison, we show also the cluster mass segregation time (grey). The shaded areas embrace the boundaries assuming COs with masses $m_{CO} = 5 - 50$ M$_\odot$.

## 4. Early Black Hole Dynamics in Nuclear Clusters and Galactic Nuclei

After the NC's assembly, its stars and COs will inevitably undergo mass segregation, which can be described in terms of the "individual" dynamical friction time of each CO or massive star from Equation (4):

$$t_{df} = 1680\text{yr} \left(\frac{R_{NC}}{2\text{pc}}\right)^{3/2} \left(\frac{M_{NC}}{2.5 \times 10^7 \text{ M}_\odot}\right)^{-1/2} \left(\frac{M_{NC}}{m_{BH}}\right)^{0.67} \left(\frac{r}{R_{NC}}\right)^{1.76} f(e,\gamma), \quad (8)$$

where, now, $M_{NC}$, $R_{NC}$, and $\gamma$ represent the NC mass, length scale, and density slope, respectively, whilst $m_{BH}$ represents the CO mass and $r$ its orbit. Note that the scaling values adopted above correspond to the Galactic NC. Note that $t_{df}$ is valid also for galaxies without a central NC—in which case, one should consider the properties of the bulge in Equation (8)—and can return a more accurate estimate of the mass segregation time in the case of galactic nuclei, which are often characterised by cuspy density profiles [2].

An important element that needs to be taken into account to estimate how many COs inhabit a galactic nucleus is related to the stellar evolution of COs progenitors, especially as concerns stellar BHs. On the one hand, stellar progenitors are somewhat heavier than their remnants; thus, dynamical friction is more effective on them. On the other hand, COs at formation can undergo a recoil, owing to the supernova (SN) kick, which can delay their orbital segregation.

The motion of a CO, or its progenitor, subjected to dynamical friction can be expressed as [278]

$$r_{\rm CO}(t) = r_{\rm CO,0}\left(1 - \beta\frac{t}{t_{\rm df}}\right)^{1/\beta},$$

(9)

with $\beta = 1.76$.

From Equation (8), a star sufficiently massive ($\gtrsim$ 18–20 M$_\odot$) starting from 0.1–1 pc from the SMBH in an MW-like nucleus will reach the centre in $t_{\rm df} \sim$ 1–80 Myr; thus, it will likely reach its last evolutionary stage while travelling through the galactic nucleus. If the newborn CO receives a natal kick, as expected for both BHs and NSs (see, e.g., [80,258,259,279]), the imparted momentum will suddenly displace the object from its orbit, or even eject it from the galactic nucleus. If the kick amplitude is smaller than the host escape velocity, the CO will eventually return toward the centre over a dynamical friction timescale.

In order to provide the reader with a simple view of how the interplay of single-star stellar evolution and dynamics can shape the evolution of COs in galactic nuclei, we exploited the B-POP population synthesis code [99], which combines stellar evolution models for single and binary BHs obtained with the MOBSE tool [280] and semi-analytic recipes to describe the motion and pairing of BHs via dynamics. Using B-POP , we considered an NC with mass $M_{\rm NC} = 2.5 \times 10^7$ M$_\odot$ and assumed that a fraction $\sim 10^{-3}$ of such mass consists of BH progenitors, assuming that the underlying mass distribution follows a standard initial mass function [255]. For each BH progenitor in the NC, we retrieved the final BH mass, the lifetime, and the SN kick. We divided the time into logarithmic bins and calculated the BH progenitor position via Equation (9). As soon as the time exceeds the *i*-th progenitor lifetime, we turned it into a BH (assuming a metallicity $Z = 0.0002$) and assigned a natal kick amplitude $v_{\rm kick}$, which was based on the stellar evolution recipes implemented in the MOBSE population synthesis tool. Given the kick, we assumed that the newborn BH will reach a maximum distance $r_{\rm max}$ in a travel time $t_{\rm tr} = r_{\rm max}/v_{\rm kick}$, and then, it comes back over a dynamical friction time. In order to simplify the visualisation of such a complex system, we present in the sketch in the left panel of Figure 4a a cartoon showing the evolution of BH progenitor stars. More quantitatively, we show in the right panel of the same figure the time evolution of the radii containing the 10%, 25%, 50%, 75%—referred to as *Lagrangian* radii—of BH mass in this simple toy model. The SN kick effect is rather small, owing to the relatively small kick amplitude compared to the NC velocity dispersion, suggesting that, in an MW-like nucleus, mass segregation is practically accomplished over a timespan of $\sim 10^{7-8}$ yr.

The population of BHs accumulated in the galactic centre will be characterised by a steeper density profile compared to normal stars and possibly by larger densities [269,281,282], a feature that can crucially affect COB formation. Once BHs have settled in, they can efficiently evacuate lighter objects (e.g., [263]), removing normal stars, but also other COs such as white dwarfs and NSs and hampering the possible formation of double-NS or NS–BH binaries.

This *shielding* operated by BH dynamics can be alleviated by the BH burning process, opening the possibility to NS–BH binary formation over a few relaxation times [130,136].

In the next section, we will describe how COs and especially BHs interact once they have gathered into the innermost regions of the host galactic centre.

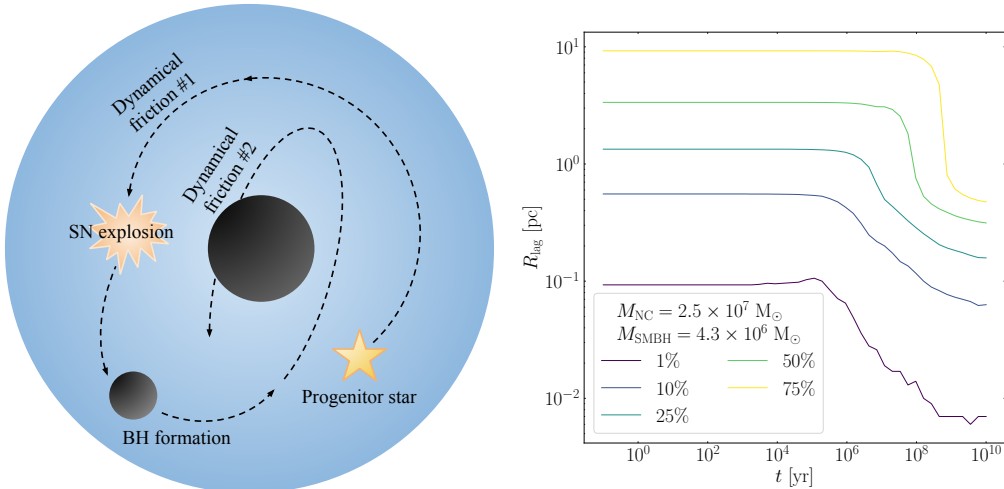

**Figure 4. Left** panel: schematic view of the orbit of a BH progenitor in a galactic nucleus. The occurrence of an SN event can impart a kick on the newborn BH and delay the segregation. **Right** panel: Lagrangian radii enclosing 10%, 25%, 50%, and 75% of the BH (or BH progenitor) mass as a function of time assuming a nucleus with mass $M_c = 2.5 \times 10^7$ M$_\odot$ and an SMBH with mass $M_{\rm SMBH} = 4.3 \times 10^6$ M$_\odot$.

## 5. Dynamical Formation of Black Hole and Compact Object Binaries in Galactic Nuclei

### 5.1. Galactic Threats: What Binaries Can Survive around a Supermassive Black Hole?

Regardless of the formation scenario, in a dense environment around an SMBH, a binary system frequently encounters other stars (e.g., [30,203,210,283–289]). This process results in a variation of the angular momentum and energy of interacting stars or COs with passing neighbours, displacing them from their position and forcing them to wander around the SMBH (e.g., [203]). When a passing star or a compact object approaches the binary with the impact parameter on the order of the binary's separation, it interacts more strongly with the closer binary member. The outcome of this interaction depends on the ratio of the binary's gravitational binding energy to the kinetic energy of the neighbouring stars. We will discuss in detail the possible outcomes of these "binary–single" interactions in Section 5.3.3.

Generally, the "resistance" of a binary against a strong interaction with another object can be quantified via the *softness* parameter (e.g., [283,290]):

$$s = \frac{E_{\rm bind}}{\langle m_* \rangle \sigma^2} \begin{cases} < 1 & \text{Soft} \\ > 1 & \text{Hard} \end{cases} \tag{10}$$

where $E_{\rm bind} = G m_1 m_2 / (2 a_{\rm bin})$ is the binary binding energy, $\langle m_* \rangle$ is the average mass of the objects (either stars or compact objects) in its vicinity, and $\sigma$ is the velocity dispersion of the environment. If the softness is larger (smaller) than unity, the binary is labelled as hard (soft), and a strong interaction will harden (soften) the binary further, a mechanism known as *Heggie's law* [283,287].

In the case of hard—or immortal [237]—binaries, interactions become rarer and more violent as they become harder and harder, until either the binary is kicked out from the environment or, in the case of COBs, GW emission kicks in and drives them toward coalescence [127]. If the hardening process involves massive stellar binaries and occurs over a timescale shorter than the typical timescale of massive star evolution, i.e., before the formation of COs, it can determine the onset of stellar collisions, which, in turn, can result in the formation of final BHs with masses larger than expected, possibly falling in the so-called pair instability mass-gap (e.g., [51,99,111,128,291]).

Soft binaries, instead, are likely subjected to disruption, either via a single strong interaction, if the perturber passes at a distance comparable to the binary semimajor axis,

or via a series of distant flybys (see [210]). In soft binary flyby interactions, the passing object increases the energy of the binary system and widens the binary, eventually leading to its *evaporation* over a timescale given by the ratio between the energy gained/lost by the binary and the rate of change of the kinetic energy, or diffusion energy coefficient, which can be expressed as [205,210,278,292]

$$
\begin{aligned}
t_{\mathrm{ev}} &= \frac{\sqrt{3}\sigma}{32\sqrt{\pi}G\rho_* \ln \Lambda a_{\mathrm{bin}}} \frac{m_{\mathrm{bin}}}{\langle m_* \rangle} \\
&= 1\,\mathrm{Gyr} \left( \frac{M_{\mathrm{SMBH}}}{4.3 \times 10^6\,\mathrm{M_\odot}} \right)^{1/2} \left( \frac{r}{0.5\,\mathrm{pc}} \right)^{-1/2} \left( \frac{\log \Lambda}{\log(15)} \right)^{-1} \\
&\times \left( \frac{\rho}{2.8 \times 10^6\,\mathrm{M_\odot pc^{-3}}} \right) \left( \frac{a_{\mathrm{bin}}}{1\mathrm{AU}} \right)^{-1} \left( \frac{m_{\mathrm{bin}}}{30\,\mathrm{M_\odot}} \right) \left( \frac{\langle m_* \rangle}{1\,\mathrm{M_\odot}} \right)^{-1}
\end{aligned}
\tag{11}
$$

and depends on the binary semimajor axis ($a_{\mathrm{bin}}$), the eccentricity ($e_{\mathrm{bin}}$), as well as on several environmental properties, such as the stellar density, $\rho_*$, the velocity dispersion, $\sigma = (GM_{\mathrm{SMBH}}/r)^{1/2}$, and the average stellar mass, $m_*$.

From the softness parameter defined above, we can determine a critical binary semimajor axis, $a_{\mathrm{hard}}$, that separates hard and soft binaries:

$$
a_{\mathrm{hard}} = \frac{2Gm_{\mathrm{bin}}}{\sigma^2} = \frac{2m_{\mathrm{bin}}}{M_{\mathrm{SMBH}}}r,
\tag{12}
$$

where the latter equality represents a valid approximation at a distance $r$ to an SMBH, where $\sigma^2 \sim M_{\mathrm{SMBH}}/r$.

Assuming a semimajor axis distribution flat in logarithmic values between 0.01 and $10^3$ AU [13,127,293] implies that $\sim$30% (70%) of binaries are hard (soft) at the Galactic Centre ($r < 0.1$ pc).

Within $r < 0.1$ pc from SgrA*, the Galactic SMBH, binaries with a semimajor axis $a_{\mathrm{bin}} > 0.29$ AU are soft; thus, their existence will be endangered by the effect of the nearby SMBH.

However, a typical equal-mass circular binary with mass $m_{\mathrm{bin}} = 30\,\mathrm{M_\odot}$ and semimajor axis $a_{\mathrm{bin}} = 0.1$ AU orbiting at a distance $r = 0.1$ pc from SgrA* evaporates in $t_{\mathrm{ev}} = 24$ Gyr; thus, even though in a soft state, the binary could still survive in the dense environment of the Galactic Centre, despite a single, strong encounter being able to disrupt it.

Having typical binary masses in the range $m_{\mathrm{bin}} = 3$–$5\,\mathrm{M_\odot}$, double-NSs can, instead, evaporate over a timescale shorter than the Hubble time, taking only $t_{\mathrm{ev}} = 2 - 4$ Gyr in the aforementioned example.

Note that the fraction of binaries with a semimajor axis smaller than $a_{\mathrm{hard}}$ in the case of a logarithmically flat distribution is given by

$$
f(< a_{\mathrm{hard}}) = \frac{\log(a_{\mathrm{hard}}/a_{\mathrm{min}})}{\log(a_{\mathrm{max}}/a_{\mathrm{min}})} = \frac{1}{\log(a_{\mathrm{max}}/a_{\mathrm{min}})} \log \left( \frac{r}{a_{\mathrm{min}}} \frac{m_{\mathrm{bin}}}{M_{\mathrm{SMBH}}} \right),
\tag{13}
$$

which implies that the fraction of hard binaries diminishes closer to the SMBH. Assuming an SMBH of mass $M_{\mathrm{SMBH}} = (1 - 4.3 - 10) \times 10^6\,\mathrm{M_\odot}$ and $a_{\mathrm{min,max}} = 0.01 - 10^3$ AU, the percentage of hard binaries drops below 10% at a distance $r_0 = (0.015 - 0.067 - 0.15)$ pc. The depletion of hard binaries toward the galactic centre has been also demonstrated via self-consistent $N-$body models of the Milky Way nucleus [282].

Figure 5 shows how the hard binary fraction varies as a function of the distance to the SMBH for the Milky Way centre, highlighting how only a tiny fraction of the most-massive binaries can still be hard within 1–10 mpc from the SMBH. At those distances, though, the hard binary separation is $a_h \sim 1\,\mathrm{R_\odot}(M_{\mathrm{SMBH}}/4.3 \times 10^6\,\mathrm{M_\odot})(r/1\,\mathrm{mpc})^{-1}$; thus, it is highly likely that a stellar binary would quickly merge.

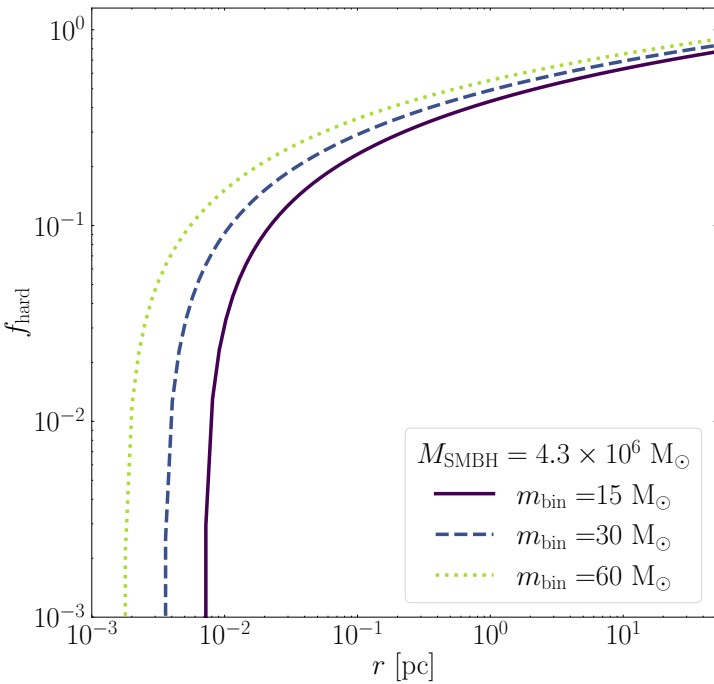

**Figure 5.** Fraction of hard binaries, normalised to the total population of binaries, as a function of the distance from an SMBH with mass $M_{\mathrm{SMBH}} = 4.3 \times 10^6$ M$_\odot$, assuming a primordial semimajor axis distribution flat in logarithmic values and limited between $10^{-2}$–$10^3$ AU.

Additionally, the evaporation process depends on the binary's eccentricity around the SMBH (e.g., [203]). In the case of soft binaries, each new interaction causes an increase of the semimajor axis with time, due to the single flyby interactions. Additionally, in the case of a soft stellar binary, mass-loss can further widen the orbit. Thus, the evaporation time of soft binaries changes with time. If we assume that the binary softens over time, we can find an upper limit on the evaporation time:

$$t_{\mathrm{ev,max}} = t_{\mathrm{ev}} S_h \tag{14}$$

where

$$S_h = \frac{s_h}{s_0} \tag{15}$$

represents the possible binary history, namely its semimajor axis evolution [290]. In particular, $s_0$ is the softness parameter calculated when the binary is observed, and $s_h$ represents the hardest possible initial configuration. In other words,

$$s_h = \min[1, s(a_{\mathrm{bin}} = R_1 + R_2)] \,, \tag{16}$$

where $R_{1,2}$ are the initial ZAMS stellar radii, assuming that the two stars were originally paired. The hardest limit taken in Equation (16) is when the binary begins as a contact binary [290].

The equations above for the evaporation timescale were derived under the assumption that the binary moves on a circular orbit about the SMBH. However, many of the stars in the Galactic Centre are in fact on an eccentric orbit (e.g., [37–39,44,206,207,294–296]). A binary on an eccentric orbit may pass through denser, inner regions of the Galactic Centre, unlike a binary on a circular orbit. Therefore, an extra term should be introduced (see for the full derivation [203]), namely

$$t_{\mathrm{ev,Ecc}} = t_{\mathrm{ev}} f(e_{\mathrm{SMBH}}) \tag{17}$$

$$
\begin{aligned}
f(e_{\mathrm{SMBH}}) \quad = \quad & \frac{(1 - e_{\mathrm{SMBH}})^{\alpha + \frac{1}{2}}}{2} {}_2F_1\left(\frac{1}{2}, -\frac{1}{2} - \alpha; 1; \frac{2e_{\mathrm{SMBH}}}{e_{\mathrm{SMBH}} - 1}\right) \\
+ \quad & \frac{(1 + e_{\mathrm{SMBH}})^{\alpha + \frac{1}{2}}}{2} {}_2F_1\left(\frac{1}{2}, -\frac{1}{2} - \alpha; 1; \frac{2e_{\mathrm{SMBH}}}{e_{\mathrm{SMBH}} + 1}\right),
\end{aligned} \tag{18}
$$

where ${}_2F_1$ is the hypergeometric function.

Binary hardening and softening processes are highly sensitive to the underlining density profile of the surrounding galaxy or NC (e.g., [48,203]), with cusp-like density profiles leading to more expedited hardening and softening processes.

In the next section, we will discuss in detail what dynamical processes intervene in the formation of binaries in galactic nuclei.

*5.2. Moving through a Swarm: Orbital Evolution of Compact Binaries in Galactic Nuclei*

As we discussed in the previous section, binaries are more likely to orbit in an outer layer of the galactic nucleus, where the effect of the central SMBH is less disruptive.

This has two crucial implications on the binary dynamics. On the one hand, the binary will drift toward the galactic centre owing to dynamical friction. On the other hand, while migrating, the binary will sweep through regions of increasing density and velocity dispersion. This will affect the boundary between hard–soft binaries and can either boost the binary shrinkage or cause its evaporation (or ionisation).

The binary infall rate can be described via the dynamical friction timescale as

$$
\frac{\mathrm{d}r}{\mathrm{d}t} = -\frac{r}{t_{\mathrm{df}}}, \tag{19}
$$

while the binary hardening rate due to binary–single interactions is given by [127,297]

$$
\frac{\mathrm{d}a_{\mathrm{BBH}}}{\mathrm{d}t} = -H\frac{G\rho(r)}{\sigma(r)}a_{\mathrm{BBH}}^2. \tag{20}
$$

If the nucleus density is $\rho \sim r^{-\gamma}$, we can express the hardening timescale as

$$
t_{\mathrm{hard}} \simeq \left(\frac{1}{a_{\mathrm{BBH}}^2}\frac{\mathrm{d}a_{\mathrm{BBH}}}{\mathrm{d}t}\right)^{-1} \propto r^{\gamma - 1/2}M_{\mathrm{SMBH}}^{1/2}; \tag{21}
$$

thus, the heavier the SMBH, the smaller the hardening and the steeper the density profile are; for $\gamma > 1/2$, the larger the hardening is as the binary migrates inward. Note that values of $\gamma < 1/2$ produce an unphysical distribution of energies for a matter distribution around an SMBH [298] and would imply that the hardening drops closer to the SMBH.

The fact that the hardening process proceeds at an increasing pace might have crucial consequences on the late evolution of the binary. For example, also the hard binary separation changes coming closer to the SMBH; thus, despite the increasing hardening, a binary could still become soft in some regions of the nucleus. Figure 6 shows the evolution of the semimajor axis normalised to the hard binary separation ($a_{\mathrm{BBH}}/a_{\mathrm{hard}}$) and distance to the SMBH ($r/r_0$) for a BBH with mass $M_{\mathrm{BBH}} = 30\,\mathrm{M}_\odot$ and different values of $a_{\mathrm{BBH}}$.

It is quite evident that, depending on the binary's properties and its initial position within the nucleus, as the binary comes closer to the SMBH, it can become softer and softer, and from that point, the evaporation time will represent a rough estimate of the binary's lifetime. Note that the $a_{\mathrm{BBH}}/a_{\mathrm{hard}}$ increase is only due to the environment, which becomes denser and hotter.

Moreover, as the binary comes closer to the SMBH, the increasing tidal field can exert important effects on the binary's evolution, which we review in the next section.

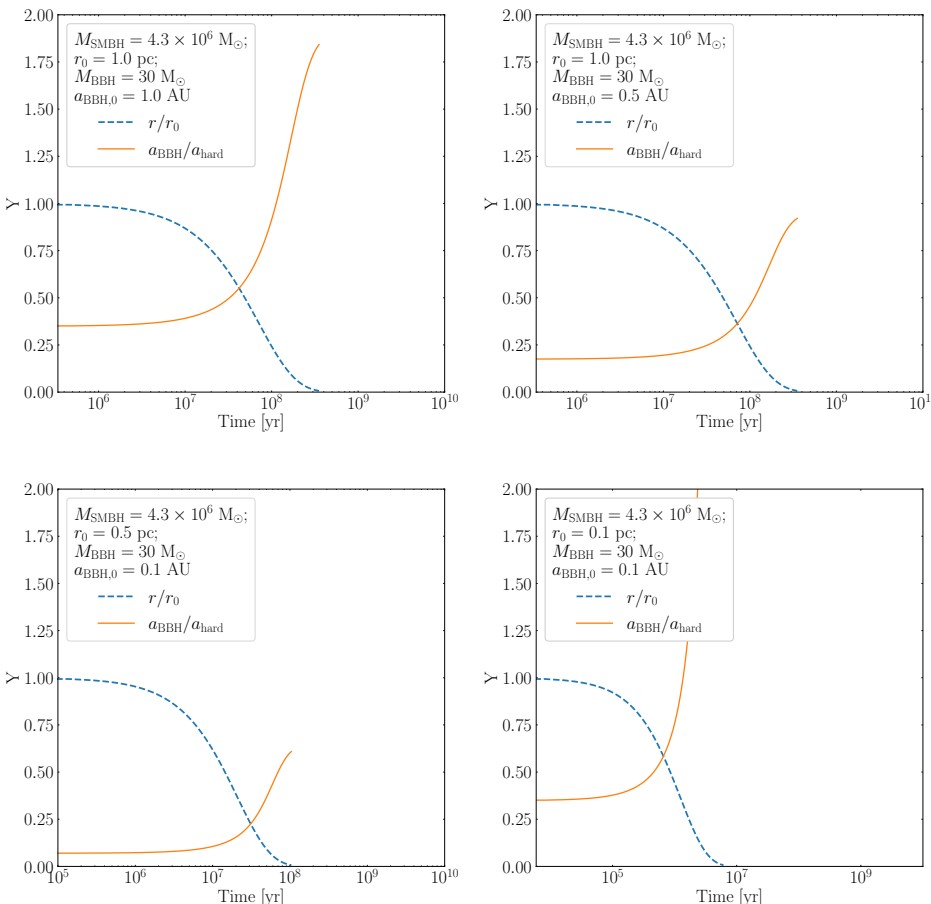

**Figure 6.** Time evolution of the BBH semimajor axis normalised to the *local* hard binary separation (orange straight line) and the BBH position within the cluster normalised to the initial position (blue dashed line). We assumed an SMBH with mass $M_{\mathrm{SMBH}} = 4.3 \times 10^6$ M$_\odot$ and a BBH with mass $M_{\mathrm{BBH}} = 30$ M$_\odot$.

*5.3. Multiple Encounters Make Bound Pairs: How Dynamical Processes Aid Binary Formation in Galactic Nuclei*

The accumulation of BHs toward the cluster centre will maximise the level of contraction of the cluster central regions, a process known as core collapse, which ideally drives the central density to grow up to infinity. The density increase onsets the formation of binaries and multiple systems that strongly interact with each other and with single objects. Binaries thus act as "heating sources" for the cluster nucleus, efficiently transferring energy to lighter single and multiple stars and ejecting them from the inner cluster regions. As a consequence, the cluster centre expands, causing a reduction of the density and velocity dispersion, which leads to a significant reduction of the interaction rate, reversing the core collapse process. The combination of mass segregation and core collapse results in a different density distribution for COs, steep and more concentrated, and stars, much shallower and sparse (see, e.g., [269,282,299]).

The energy exchange promoted by mass segregation translates into a redistribution of energy among the objects with different masses, which can trigger the formation of bound binaries via few-body interactions.

The earliest studies on the formation of binaries in dense stellar environments date back to the early 1960s, when the first Fokker–Planck, Monte Carlo, and *N*-body models were developed (e.g., [11,283,300–309]).

These pioneering experiments were developed to understand the complex evolution of star clusters and shed light on the fundamental dynamics regulating the formation and

disruption of binaries in star clusters and the feedback that binaries have on the whole cluster's evolution.

As we will see in the next sections, such fundamental dynamics can be generally dissected into several processes, namely single–single GW captures and three-body, binary–single, and binary–binary scatterings.

### 5.3.1. GW Capture Binary Formation

Aside from three-body interactions, a binary can form also via GW bremsstrahlung during a single–single interaction, provided that the two objects come sufficiently close. The maximum impact parameter below which two BHs pair can be calculated by equalling the potential energy between the interacting objects and the energy lost to GWs during the closest passage and can be expressed roughly as [264,310]

$$b_{\mathrm{bnd}} = 2.4 \mathrm{R}_\odot \left( \frac{\sigma}{100 \mathrm{km\ s}^{-1}} \right)^{-9/7} \left( \frac{m_1 + m_2}{10\ \mathrm{M}_\odot} \right) \left( \frac{\eta}{0.25} \right), \tag{22}$$

where $\eta = m_1 m_2 / (m_1 + m_2)^2$ is the asymmetric mass ratio. For an MW-like nucleus, assuming a typical BH number density of $10^6\ \mathrm{pc}^{-3}$ and velocity dispersion $\sigma \sim 100\ \mathrm{km\ s}^{-1}$, this condition implies around 0.01–0.1 binary captures per Gyr (e.g., [264,311]). The timescale for these interactions can be written as a function of the SMBH:

$$t_{\mathrm{cap}} = (n\sigma\pi b_{\mathrm{bnd}}^2)^{-1} = 3.5 \times 10^{14} yr \left( \frac{n}{10^6\ \mathrm{pc}^{-3}} \right)^{-1} \tag{23}$$

$$\times \left( \frac{M_{\mathrm{SMBH}}}{4.3 \times 10^6\ \mathrm{M}_\odot} \right)^{11/14} \left( \frac{r_0}{0.22\ \mathrm{pc}} \right)^{-11/14} \left( \frac{r}{r_0} \right)^{-11/14+\gamma},$$

where we assumed $\sigma = (M_{\mathrm{SMBH}}/r)^{1/2}$ and $n = n_0 (r/r_0)^\gamma$. Figure 7 shows the GW capture timescale for different values of the number density of BHs in the nucleus and the SMBH mass. It appears evident how this process can operate quickly close to the most-dense nuclei and the most-massive SMBHs.

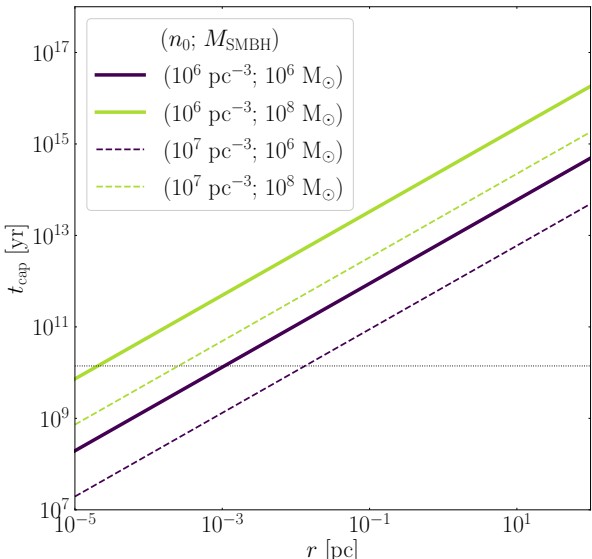

**Figure 7.** Timescale for gravitational wave capture assuming a black hole mass $m_{\mathrm{BH}} = 15\ \mathrm{M}_\odot$ as a function of the distance from the SMBH and for different values of the SMBH mass and BH number density.

At formation, the pericentre of a GW capture binary is typically $r_p \lesssim 7.4 \times 10^{-3} \mathrm{R}_\odot (\sigma/100 \mathrm{km\ s}^{-1})^{-4/7}$; thus, these binaries are characterised by extremely

short merging times, the order of minutes to hours, and their merger rate will directly follow the binary formation rate, which is expected to be in the range of 0.2–150 Gyr$^{-1}$ [264].

As we will discuss further in Section 8, among all the channels for BBH formation in galactic nuclei, the GW capture enables the production of highly eccentric sources ($e > 0.1$) in the typical frequency band where the sensitivity of ground-based interferometers such as LVC peaks, i.e., f > 1–10 Hz.

### 5.3.2. Three-Body Binary Formation

By definition, a *three-body* interaction involves three unbound objects scattering onto each other. Statistically speaking, this type of interaction leads to the ejection of the least-massive object and the formation of a bound pair. The formation rate for three-body binaries can be obtained by equalling the rate at which new binaries form and soft binaries are ionised via multiple or single interaction and can be expressed as [11,312]

$$\frac{dn_{3bb}}{dt} = \alpha(Gm)^5 n^3 \sigma^{-9},$$ (24)

where $\alpha$ is a normalisation constant, $m$ is the average mass of the interacting bodies, $n$ is the environment number density, and $\sigma$ is its velocity dispersion. The timescale associated with this process, $t_{3bb} \sim m^{-5}n^{-2}\sigma^9$, immediately highlights how crucial the environment is, and its evolution, in determining binary formation via three-body interactions.

The binary formation rate above was derived under the assumption that only hard binaries are "immortal" [283]. However, in environments with a particularly large velocity dispersion, thus a small hard binary separation limit, soft binaries can harden via GW emission and cross the soft–hard boundary, provided that the hardening time is shorter than the evaporation time. This mechanism to harden soft binaries works for particularly dense nuclei where encounters with a relative velocity close to the speed of light ($v_{\rm rel} \gtrsim 0.3c$) become possible [313].

At a distance $r$ from an SMBH, the velocity dispersion is Keplerian, $\sigma^2 \propto M_{\rm SMBH}/r$, whilst the matter density is expected to follow a power-law [299,314–319], $n \propto r^\gamma$, with $\gamma = -3/2 - m/4m_{\rm max}$ and $m(m_{\rm max})$ the average (maximum) stellar mass [318,320]. For a monochromatic mass spectrum, the matter density around an SMBH follows the so-called Bahcall–Wolf cusp, with slope $-7/4$ [314]. In a multimass system, stellar BHs, which are the heaviest objects in a stellar ensemble, are expected to distribute according to a Bahcall–Wolf cusp or even a steeper one [264,265,299,319,320], whilst lighter stars follow a shallower density distribution [266,281,282,321].

The energy transfer from the heaviest to the lightest object progressively leads to a decrease of the heavy object velocity dispersion owing to the equipartition principle. Mass-segregated BHs are characterised by a velocity dispersion $\sigma_{\rm BH}/\sigma_* \sim 1/\zeta(m_{\rm BH}/m_*)^{-\eta}$, with $\zeta \leq 1$. In the ideal case of "perfect equipartition", $\eta = 0.5$ [209]; however, simulations have shown that equipartition is hard to reach in dense clusters, even in the presence of a central massive object, as shown for globular cluster models hosting a central IMBH, which suggest $\eta = 0.08 - 0.15$ [322].

Given the dependencies above, the three-body timescale can be re-written as

$$t_{3bb} = (Gm_{\rm BH})^{-5} n_{\rm BH}^{-2} \sigma_{\rm BH}^9$$ (25)

$$= (Gm_{\rm BH})^{-5} \left(\frac{\rho_0}{m_*}\right)^{-2} \left(\frac{GM_{\rm SMBH}}{r_0}\right)^{9/2} \left(\frac{m_*}{m_{\rm BH}}\right)^{9/2} \left(\frac{r}{r_0}\right)^{2\gamma - 9/2}$$

$$= 12.5 \, {\rm Gyr} \left(\frac{m_{\rm BH}}{30 \, {\rm M_\odot}}\right)^{-19/2} \left(\frac{\rho_0}{2.8 \times 10^6 \, {\rm M_\odot \, pc^{-3}}}\right)^{-2} \left(\frac{M_{\rm SMBH}}{4.3 \times 10^6 \, {\rm M_\odot}}\right)^{9/2}$$

$$\times \left(\frac{r_0}{0.22 \, {\rm pc}}\right)^{-9/2} \left(\frac{m_*}{0.5 \, {\rm M_\odot}}\right)^{13/2} \left(\frac{r}{r_0}\right)^{2\gamma - 9/2},$$

where we used the fact that $n = \rho_0/m_*(r/r_0)^\gamma$ and $\sigma^2_{BH} = (m_*/m_{BH})GM_{SMBH}/r$. Note that the Galactic NC is characterised by $\rho_0 \sim (2.8 \pm 1.3) \times 10^6 \, M_\odot pc^{-3}(r/r_0)^{-\gamma}$, with $\gamma = 1.2(1.75)$ inside (outside) the inner $r_0 = 0.22$ pc [39,323].

Note that for NSs, the three-body interaction time becomes incredibly long, owing to the steep dependence on the CO mass. The $t_{3bb}$ time becomes shorter than the Hubble time only if we consider nuclei with considerably lighter SMBH, $\sim 10^5 \, M_\odot$, and in the outermost ($r > 5$ pc) regions of the NC.

Under the rather simplistic assumption that the NC's properties do not vary significantly over time, Equation (24) can be integrated over time to obtain the following, time-dependent, expression of the binary fraction (for a full derivation, see [278]):

$$f_{BBH} = \frac{1}{2}\left(1 - \frac{1}{\sqrt{1 + t/t_{3bb}}}\right). \tag{26}$$

Figure 8 shows the BH binary fraction (over the total population of BHs) as a function of the distance to the Galactic Centre after a time t = 0.1–10 Gyr, assuming an average BH mass $m_{BH} = 15 \, M_\odot$. The figure makes clear that a pure three-body formation process is highly inefficient in the innermost Galactic regions, owing to the large velocity dispersion, which hampers three-body interactions. Nonetheless, it also highlights that, at distances 1–10 pc, the BBH fraction attains values $f_{BBH} \sim 10^{-3} - 10^{-2}$, which could imply the formation of a few tens of BBHs over a 10 Gyr timespan, but how many? Let us assume that the inner pc in the Milky Way contains around $N_{BH}(< 1pc) = 20,000$ BHs [53] and that the BH density distribution at the Galactic Centre is represented by a simple broken power-law [324], with a total number of BHs $N_{BH} = 10^{-3}N_{NC}$, a scale radius of $a = 0.1$ pc, and a density slope $\gamma = 2$ (this choice of parameters returns a number of BHs inside 1 pc similar to the value inferred from observations of the Galactic Centre). It is possible to show that, at a distance 1–10 pc from the Galactic Centre, we expect $O(10^3)$ BHs; thus, the number of binaries formed in the Galactic NC in 10 Gyr solely via three-body scattering would be $N_{BBH} \sim 2$–20. A way to increase the number of BBHs is via exchange in binary–single scatterings involving a binary star and a single BH (e.g., [127,278]).

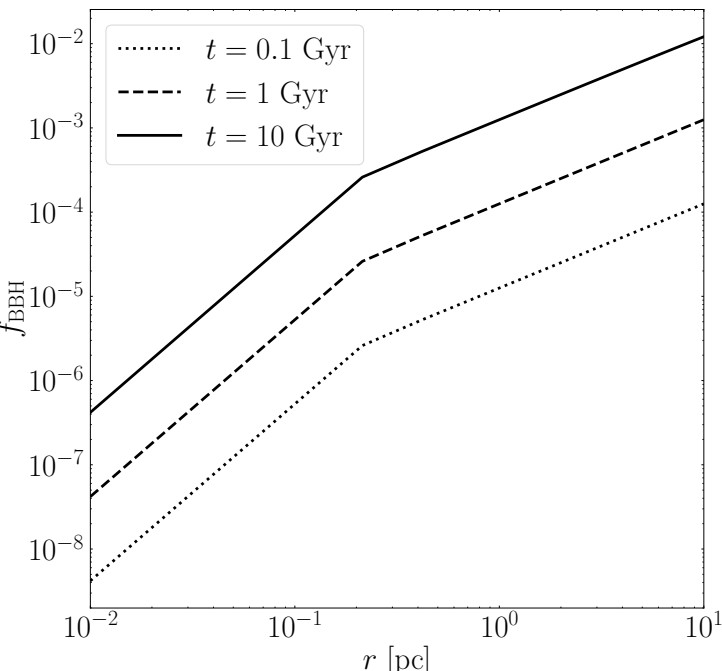

**Figure 8.** Fraction of binary black holes as a function of the distance to the Galactic Centre after an evolutionary time of 0.1 Gyr (dotted line), 1 Gyr (dashed line), and 10 Gyr (straight line).

For the NC in the Milky Way and a CO average mass of $m_{\text{BH}} = 30\,\text{M}_\odot$, the three-body timescale is shorter than the dynamical friction timescale at distances $r > 3.8$ pc, thus suggesting that binaries containing at least one BH could form in the outer NC regions.

From the equation above, for a Bahcall–Wolf cusp—$\gamma = -7/4$—the three-body interaction time increases toward the SMBH as $t_{\text{3bb}} \propto 1/r$.

Equation (25) implies that three-body scatterings are highly suppressed for objects lighter than $30\,\text{M}_\odot$; thus, the formation of stellar binaries in such extreme environments must resort to other mechanisms to sustain their binary population or need to harbour a substantial fraction of "primordial" hard binaries.

It must be also noted that the steep dependence on the density and velocity dispersion makes the three-body time estimates extremely susceptible to the choice of the initial conditions, as suggested by Figure 9, which shows how $t_{\text{3bb}}$ varies at varying SMBH and NC masses and the distance to the galactic centre. This is owed to the fact that $t_{\text{3bb}} \propto M_{\text{NC}}^{-2} M_{\text{SMBH}}^{9/2}$; thus, reducing the SMBH mass by five-times causes a reduction of the three-body time by a factor of $\sim$1400. For the sake of comparison, note that, for BHs with mass $\sim10\,\text{M}_\odot$ in a typical globular cluster, with density $n \sim 10^4$ pc$^{-3}$ and velocity dispersion $\sigma = 5$–10 km s$^{-1}$, the three-body timescale reduces to $t_{\text{3bb}} \sim 0.004$–2 Gyr.

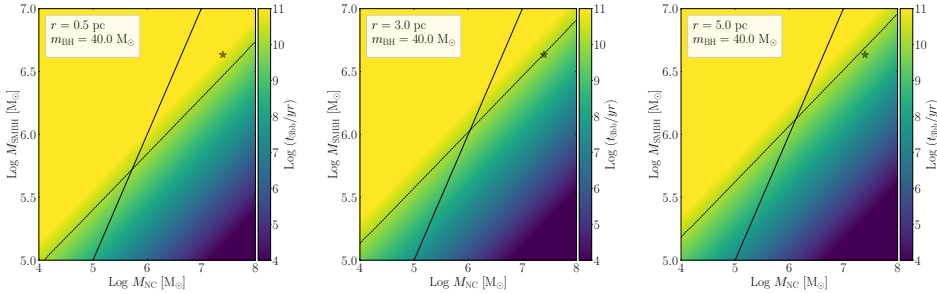

**Figure 9.** Three-body timescale as a function of the NC (x-axis) and SMBH (y-axis) mass. We assumed three equal-mass BHs ($m_{\text{BH}} = 40\,\text{M}_\odot$) orbiting at $r = 0.5$–0.3–5 pc from the SMBH (from left to right, respectively). The star marks typical values in the Milky Way. The straight line marks the case $M_{\text{NC}} = M_{\text{SMBH}}$, whilst the dotted line separates the region with a $t_{\text{3bb}}$ smaller (below the line) or larger (above the line) than the Hubble time.

5.3.3. Binary–Single Scatterings

The presence of even a few binaries in the nucleus, either primordial or formed dynamically, will lead to a series of binary–single interactions with other members.

The cross-section of a strong encounter between a binary with mass $m_{\text{bin}}$ and semimajor axis $a$ and a perturber $m_{\text{p}}$ from the binary can be expressed as

$$\Sigma = \pi a^2 (1-e)^2 \left[ 1 + \frac{2G(m_{\text{bin}} + m_{\text{p}})}{3\sigma^2 a (1-e)} \right]. \tag{27}$$

The number of interactions per time unit that the binary will undergo while moving with velocity $\sigma$ in an environment with stellar density $n$ is $\mathrm{d}n \simeq n\Sigma\sigma\mathrm{d}t$. The associated timescale of this process, $t_{\text{bs}} = (n\Sigma\sigma)^{-1}$, can be conveniently written as [13,127]

$$t_{\text{bs}} = 3.2 \times 10^9 \text{ yr} \left( \frac{n}{10^6 \text{ pc}^{-3}} \right)^{-1} \left( \frac{f_{\text{bin}}}{0.01} \right)^{-1} \tag{28}$$

$$\times \left( \frac{r}{2 \text{ pc}} \right)^{-1/2} \left( \frac{M_{\text{SMBH}}}{4.3 \times 10^6 \text{ M}_\odot} \right)^{1/2} \left( \frac{m_{\text{bin}} + m_p}{30 \text{ M}_\odot} \right)^{-1} \left( \frac{a}{1\text{AU}} \right)^{-1},$$

where $f_{\text{bin}}$ represents the binary fraction.

If the perturber is a BH, whilst the binary contains stars, the interaction will take place proportionally to the fraction of binaries present in the cluster. In such a case, the binary will likely acquire the BH if it is more massive than at least one component.

In this case, Equation (28) remains valid even in the case of binary–binary interactions, provided that the perturber mass is replaced with the typical binary mass. Binary–binary interactions represent a viable way to produce CO triples. On the one hand, triples can undergo secular evolution, with the least-bound object—or outermost—possibly impinging EKL oscillations onto the most-bound pair, eventually driving it to coalescence [159,163,325]. On the other hand, triples formed out of binary–binary interactions can undergo a chaotic unstable evolution, which, in some cases, can trigger the formation of highly eccentric merging binaries [159,163]. Figure 10 shows one example of BBH–BBH strong scattering from numerical simulations described in [163]. The plot highlights how, after the encounter between the two binaries, one of the BHs—the lightest—is ejected away and an unstable triple forms. The triple eventually breaks up and leads to the formation of a tight binary, which merges after $10^2$ yr (see Figure 6 in [163]).

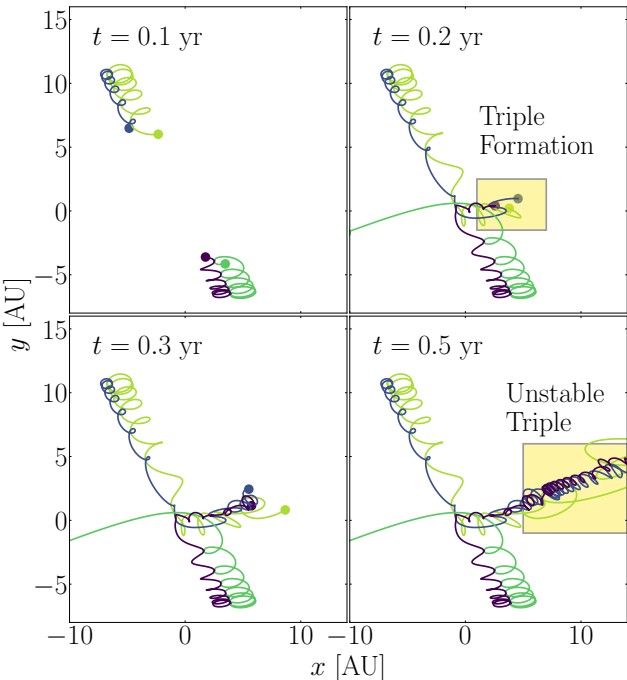

**Figure 10.** Binary–binary interaction involving four BHs in a dense star cluster. After the close encounter, one BH is ejected and the remaining three BHs form an unstable triple. Taken from Figure 2 in Arca Sedda et al., A&A, 2021, 650, A189, reproduced with permission ©ESO [163].

The presence of an SMBH in the galactic centre can significantly influence triple evolution, tidally limiting them and enabling only the formation of triples whose maximum size remains within the triple Roche lobe [326], possibly boosting the COB merger rate owing to a more efficient development of the EKL mechanism [327].

If, instead, the BH is already in a binary, it can interact with all the stars in the nucleus and the actual binary–single scattering time becomes $t_{bs,BBH} \sim f_{bin} t_{bs}$; thus, BHs in binaries can undergo binary–single interactions at a rate up to 100-times larger than single BHs.

If NSs or BHs are paired in a binary already, either because of previous dynamical processes or owing to primordial stellar evolution, it has been shown that binary–single interactions are particularly efficient at producing merging NS–BH binaries [130,132,133,136], especially in dense galactic nuclei [130]. The typical properties of dynamical NS–BH mergers formed this way, or at least of a sub-population of them, differ significantly from the observed properties of NS–BH binaries formed in isolation, permitting clearly identifying their origin in GW observation ([130,136]; however, see also Section 8). Figure 11 shows the

mass spectrum of merging NS–BH binaries formed via binary–single interactions in dense clusters with velocity dispersion in the 10–100 km/s range, for two different values of the stellar metallicity. The distribution takes into account observational biases, in particular the fact that LIGO-Virgo detectors can access a larger volume for nearly equal-mass binaries with large masses in comparison to low-mass or highly asymmetric binaries [136,328].

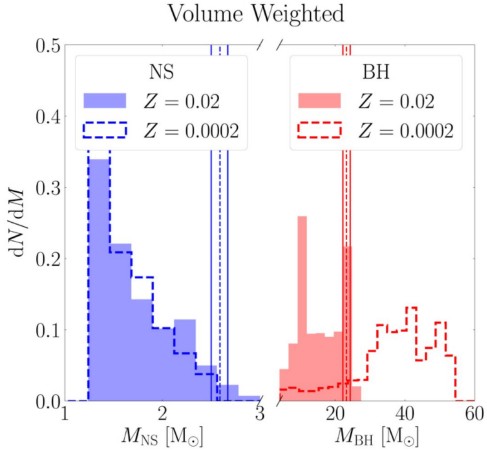

**Figure 11.** Merging NS–BH binaries formed via binary–single encounters in dense clusters and corrected for ground-based GW detectors. Vertical lines correspond to the median value (dashed line) and 95% confidence level boundaries (straight lines) of the mass of GW190814 source components. Taken from Figure 4 in Arca Sedda, ApJL, 2021, 908, L38 ©AAS [136]. Reproduced with permission.

The typical outcome of a binary–single encounter depends on the ratio between the energy transferred during the encounter, which is roughly given by $\Delta E_{bs} \sim (r_{p,e}/a)^{3/2}$ [283,329], where $r_{p,e}$ represents the minimum binary–perturber separation, and the kinetic energy calculated in the centre of mass of the triple, i.e., [329]

$$\Delta E_c = v_\infty^2 \frac{m_p m_{bin}}{2(m_p + m_{bin})},\qquad(29)$$

with $v_\infty$ the velocity at infinity of the perturber in the binary reference frame. If the energy transferred is small, i.e., $\Delta E_{bs} < \Delta E_c$, the binary has two possible fates, depending on its binding energy $E_{bin}$:

i     If $\Delta E_{bs} < E_{bin}$, the binary will harden or soften depending on the environment;

ii    If $\Delta E_{bs} > E_{bin}$, the binary will exchange one component, most likely the least-massive one, if the perturber is heavier than the binary or its components, i.e., $m_p > (m_1 + m_2)$ or at least $m_p > m_{1,2}$.

If the binary preserves its components, the perturber will recede to infinity with velocity $v_p < v_\infty$; otherwise, the former binary component with mass $m_{exc}$ will escape with velocity $v_p < \sqrt{(m_p/m_{exc})}\,v_\infty$.

In the case of a high-energy transfer, $\Delta E_{bs} > \Delta E_c$, the outcome of the scattering is a bit more complex, but can be determined statistically by comparing the interaction velocity $v_\infty$ and the critical velocity value above which the perturber can promptly ionise the binary, defined as [330]

$$v_c^2 = \frac{G m_1 m_2 (m_p + m_{bin})}{a m_p m_{bin}}.\qquad(30)$$

In such a way, the possible final states can be categorised as follows:

i     If $v_\infty < v_c$, the perturber cannot recede to infinity and the three bodies undergo a resonant interaction that can culminate in the exchange of one binary component if:

     –    $\Delta E_{bs} > E_{bin}$;

     –    $\Delta E_{bs} < E_{bin}$ and $m_p > m_{bin}$.

Either way, the perturber or the exchanged component recedes to infinity and possibly leaves the host system;

ii    If $v_\infty < v_c$, $\Delta E_{bs} < E_{bin}$, and $m_p < m_{bin}$, the system undergoes a resonant interaction, which generally leads to the ejection of the lighter component;

iii   If $v_\infty > v_c$, the binary undergoes:

    –   Component exchange if $\Delta E_{bs} < E_{bin}$;

    –   Ionisation if $\Delta E_{bs} > E_{bin}$.

For a BH population with equal mass $m_{BH}$ and a binary population with semimajor axis $a_{BBH} = 1$ AU, the critical velocity is $v_c \sim 50$ km s$^{-1}$. The speed of the encounters around an SMBH, instead, is $\sigma_{SMBH} = 65$km s$^{-1}(M_{SMBH}/10^6 \, M_\odot)^{1/2}(r/1\text{pc})^{-1/2}$.

This implies that the binary–single interactions inside the SMBH's influence radius will be characterised mostly by $v_\infty \sim \sigma_{SMBH} > v_c$.

Depending on the masses involved and the semimajor axis of the binary, it can be shown that the energy transferred in a binary–single encounter rapidly drops when $\sigma/v_c > k$, where $k \sim 0.1$–1, whilst the critical energy transfer $\Delta E_c$ increases with $(\sigma/v_c)^2$. Thus, it seems reasonable to expect that, in the closest vicinity of an SMBH, binary–single encounters tend to favour small energy transfer. Since the binary in these encounters is likely hard, thus $E_{bin} \propto \sigma^2$, binary–single interactions close to an SMBH might result statistically in the binary hardening, i.e., the cases $\Delta E_{bs} < \Delta E_c$ and $\Delta E_{bs} < E_{bin}$. In other words, component exchange and binary ionisation might be significantly suppressed in galactic nuclei unless the binary semimajor axis is close to the hard–soft separation value (see, e.g., [278]). This simple line of thought finds support in recent scattering simulations tailored to binary–single scattering around an MW-like SMBH (see, e.g., [326]). After the interaction, the binary will recoil at a velocity $v_{rec} = k\sigma$ [127], with $k \simeq 0.3$ for a monochromatic BH mass spectrum.

## 6. Secular Dynamical Effects on Binary Evolution around a Supermassive Black Hole

*6.1. Secular Perturbations on Black Hole Binaries in Galactic Nuclei: The Impact of a Supermassive Black Hole*

While the effects of binary interactions have been shown to play a crucial role in the global dynamical evolution of dense systems such as globular clusters, only recently have the effects of binaries in NCs been investigated. Within the vicinity of an SMBH, the members of a stable binary have a tighter orbital configuration than the orbit of their mutual centre of mass around the SMBH. In such a system, gravitational perturbations from the SMBH can induce large eccentricities on the binary orbit, which can cause the binary members to merge. Below, we outline the relevant physical processes that characterise the COB–SMBH co-evolution.

### Three-Body Newtonian Limit

In the three-body approximation, dynamical stability requires that the system has either a circular, concentric, coplanar orbits, or hierarchical configuration, in which the inner binary, with semimajor axis $a_1$, is orbiting a third body (in this case, the SMBH) on a much wider orbit, the outer binary, with semimajor axis $a_2$. In such hierarchical configurations, it is possible to apply the so-called secular approximation. In essence, this means that the orbital period can be averaged; thus, the interactions between two non-resonant orbits are equivalent to treating the orbits as massive "wires", where the line-density is inversely proportional to the orbital velocity. Under this approximation, the gravitational potential is expanded in the ratio of orbital separations—which, in this approximation, remains constant, since $(a_1/a_2) \ll 1$ (e.g., [14–16,21,331]).

The Hamiltonian of such a system can be written as

$$\mathcal{H} = -\frac{Gm_1 m_2}{2a_1} - \frac{GM_{SMBH}(m_1 + m_2)}{2a_2} - \mathcal{R}_{pert}\,, \tag{31}$$

where the first two parts represent the Keplerian energies of the two orbits and $\mathcal{R}_{\mathrm{pert}}$ is the perturbation function [332]. The perturbation function can be written as a function of the orbital (Delaunay's) elements, the arguments of periastron, $\omega_1$ and $\omega_2$, the longitudes of ascending nodes, $\Omega_1$ and $\Omega_2$, and the mean anomalies $\mu_1$ and $\mu_2$, for the inner and outer orbits, respectively. In particular,

$$\mathcal{R}_{\mathrm{pert}} = \frac{G}{a_2} \sum_{n=2}^{\infty} \left(\frac{a_1}{a_2}\right)^n \mu_n \left(\frac{r_1}{a_1}\right)^n \left(\frac{r_2}{a_2}\right)^{n+1} P_n(\cos\psi), \tag{32}$$

where $r_1$ ($r_2$) is the position radius of the inner (outer) orbit, $P_n$ are the Legendre polynomials, and $\psi$ is the 3D angle between the position vectors of the inner and outer orbits.

The nominal procedure in such a system is to perform a canonical transformation to a set of coordinates that eliminates the short-period angles of the inner and outer orbits. This transformation is known as the von Zeipel transformation; see for details Appendix B in [332].

The resulting (often called double-averaged) Hamiltonian is, thus, a multiple expansion of $a_1/a_2$. The lowest order of approximation (proportional to $(a_1/a_2)^2$) is called the quadrupole level and corresponds to $n = 2$ in Equation (32).

In early studies of high-inclination, hierarchical, secular perturbations [14,15], the outer orbit (in this case, about the SMBH) was assumed to be circular, with one of the inner binary members being a test (massless) particle. In this scenario, the component of the inner orbit's angular momentum along the z-axis (set to be the total angular momentum) is conserved—known as Kozai's integral of motion—and the lowest order of the approximation, the quadrupole approximation, is valid.

In this regime, the conservation of the vertical angular momentum and energy implies that the eccentricity and mutual inclination vary along the orbit. An illustrative case is represented by an initially circular binary inclined by an angle $i_0$ with respect to the binary–SMBH orbital plane, in which case the maximum eccentricity is given by (e.g., [19,188,333])

$$e_{\mathrm{max}} = \sqrt{1 - \frac{5}{3}\cos^2 i_0}. \tag{33}$$

The eccentricity excitation is triggered above a minimum value of the initial inclination, given by $\cos i_{\mathrm{min}} = \pm\sqrt{3/5}$, thus implying that only orbits having $39.2° < i_{\mathrm{min}} < 140.77°$ can undergo eccentricity variations. The timescale for KL oscillations to develop is (e.g., [334])

$$
\begin{aligned}
t_{\mathrm{KL}} =\ & \frac{16}{30\pi} \frac{m_1 + m_2 + m_3}{m_3} \frac{P_2^2}{P_1} (1 - e_2^2)^{3/2}, \\
=\ & 40\,\mathrm{Myr} \left[ \left(\frac{m_1 + m_2}{30\,\mathrm{M}_\odot}\right) \left(\frac{m_3}{10^6\,\mathrm{M}_\odot}\right)^{-1} + 1 \right] \left(\frac{a_2}{0.1\,\mathrm{pc}}\right)^3 \left(\frac{a_1}{0.1\,\mathrm{AU}}\right)^{-3/2} (1 - e_2^2)^{3/2},
\end{aligned} \tag{34}
$$

with $P_i = \sqrt{a_i^3/(Gm_{\mathrm{bin,i}})}$ the inner ($i = 1$) or outer ($i = 2$) binary orbital period.

About a decade ago, it was shown that relaxing either one of these assumptions leads to qualitatively different dynamical behaviours (already at the quadrupole level). Considering systems beyond the test particle approximation, or a circular orbit, requires higher terms in the Hamiltonian, called the octupole-level of approximation, proportional to $(a_1/a_2)^3$, i.e., $n = 3$ in Equation (32) (e.g., [16–18,331]). In the past few years, the octupole-level approximation was proven to have an important role in the evolution of many triple configurations, for different astrophysical settings (e.g., [19,20,188,205,292,332,333,335–345]).

In the octupole level of approximation, the inner orbit eccentricity can reach extremely high values [17,332,345,346]. This process was coined as the eccentric Kozai–Lidov (EKL) mechanism [21]. The hallmark of the EKL mechanism results in large eccentricity peaks

and chaotic behaviour and also flips the inclinations of both the inner and outer orbits from prograde to retrograde [19,333,335,346–349].

The quadrupole level of approximation relates to the low-level resonance, which causes the precession of the orbits and, thus, results in the excitation of the orbital inclination and eccentricity (e.g., [332,347–350]). The higher-level approximation results in resonances that often are overlapping. These resonances result in extreme eccentricity values, as well as flips and chaos (e.g., [19,332,333,335,346,347,349,351]). These extreme eccentricities are valuable for the mergers of compact objects. In Figure 12, left side, we show a representative example of the effect of the octupole on merging binary. In general, octupole effects shorten the timescale and, thus, have a significant effect on the merger rate [292]; see Figure 12, left panel, for example.

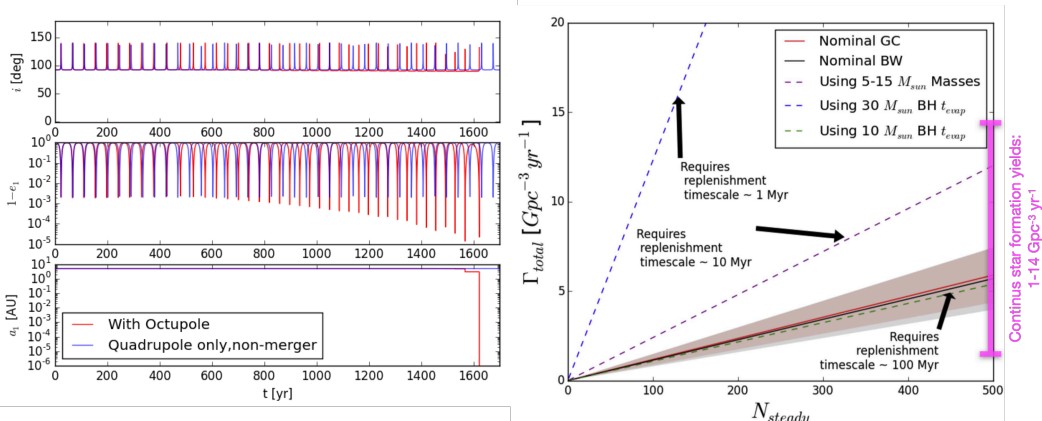

**Figure 12.** (**Left panel**) A representative example from [292], that depicts the evolution of a BBH while solving the deafening equations up to the octupole level of approximation (red) and only up to the quadrupole level of approximation (blue). As shown, the octupole-level evolution results in a merger after 1623 years, while the quadrupole-level approximation never merges. (**Right panel**) Volumetric BH merger rate as a function of the number of BBHs. This rate represents Monte Carlo results from Hoang et al, ApJ, 2018, 856(2), 140 ©AAS [292]. Reproduced with permission.

## 6.2. Other Physical Processes

Aside from two-body relaxation, which we discussed in the previous section, and the secular effect of the SMBH field on binary members, different physical and astrophysical processes can contribute to the overall evolution of the system. These include short-range forces between the binaries, as well as dynamical interaction with the neighbouring objects in NCs. Below, we briefly review these processes and their affects.

**Tidal dissipation**

Considering the full evolution of binaries (i.e., starting from main-sequence stars), requires the inclusion of tidal dissipation. As the system evolves, the inner orbit can become highly eccentric, which could, on the one hand, drive the inner binary to merge, while on the other hand, it could allow tidal forces to shrink and circularise the orbit. The latter happens if, during the evolution, the tidal precession timescale (or the general relativistic (GR) timescale) is similar to that of the lowest EKL timescale, which corresponds to the quadrupole. Thus, further eccentricity excitations are suppressed (e.g., [340,350,352]), and tides shrink the binary separation, forming a tight binary that is decoupled from the SMBH. Conversely, if the eccentricity is excited on a shorter timescale than the typical tidal (or GR) precession timescale, the orbit becomes almost radial, and the stars can cross the Roche limit. In that case, the tidal precession does not have enough time to affect the evolution.

Most studies in this field use the equilibrium tide models [353,354]. The strength of the equilibrium tide model is that it is self-consistent with the secular approach used throughout this study. Further, assuming that the stars are polytropic, this model has only one dissipation parameter for each member of the binary. Using this tidal description,

we are able to follow the precession of the spin of each star in the binary due to the stars' oblateness and tidal torques (e.g., [340]). The disadvantage of using equilibrium is that, for sufficiently large eccentricity, it tends to underestimate the efficiency of the tides compared to chaotic dynamical tides (e.g., [355,356]).

While the equilibrium tidal model seems to be roughly consistent with the qualitative behaviour of polytropes with convective envelopes, tides for radiative stars are estimated to be much weaker than for convective stars. Thus, within this framework, a different tidal model for (radiative) main-sequence and (convective) red giant stars is invoked (e.g., [357,358]). Within the context of EKL evolution, only a few studies began including the less-efficient radiative tides (e.g., [59,342,344,359,360]).

Overall, tides between the binary members largely influence the EKL evolution of the stars. They tend to shrink and circularise the orbit (e.g., [205,340]), thus hardening the binary to flyby interactions (e.g., [203,361]). During the binary stellar evolution, if the eccentricity spikes take place faster than the tides can suppress the high eccentricity values, the binary stars may merge (typically of few $\times 10 \, M_\odot$), and they are speculated to form a G2-like object (e.g., [59,205,362,363]). Surviving stars undergo a combination of EKL, tides, GR, and stellar evolution (see below).

**Stellar evolution**

Stellar evolution plays an important role in the evolution of binaries, and especially massive binaries. Specifically, it was shown that mass-loss can have a significant effect on the dynamical evolution of binaries and triples (e.g., [188,205,338,364–367]). For near-equal-mass binaries, the mass-loss associated with the AGB phase can re-trigger the EKL mechanism, either by changing the mass ratio or by expanding the semimajor axis of the inner orbit faster than that of the outer one [338,368].

Often, in the literature, the post-main-sequence evolution model is adopted from a combination of stellar evolution codes, such as BSE/SSE [154] and MESA [369], which are publicly available. Furthermore, once the binaries cross each other's Roche limit, the binary stellar evolution is often followed using COSMIC binary stellar evolutionary code [370].

The effects of supernova explosions and asymmetric and/or instantaneous mass-loss (i.e., pulse-like, instantaneous, relative to the secular timescale) can affect the orbital parameters of *massive* binaries (e.g., [371–374]) and triples (e.g., [375,376]). Sudden mass-loss can cause a rapid change of the mass ratio, but more importantly, it can change the eccentricity of the inner and outer orbit due to a supernova kick.

In particular, there are a variety of dynamical outcomes resulting from SN kicks, in binaries at the centre of galaxies. For example, they can result in a hypervelocity star (e.g., [377]) and binary candidates (e.g., [375,378]), as well as X-ray binaries [378]. Finally, GW events triggered by SN kicks can result in binary merger events and EMRIs (e.g., [375,377,378]).

**General relativity precession, first pN**

In the first post-Newtonian (pN) approximation, the inner (and outer) binary exhibits an additional precession of the nodes. The typical timescale for this mechanism can be written as [18,379]

$$t_{1\text{pN}} = \frac{a_1^{5/2} c^2 (1 - e_1^2)}{3 G^{3/2} m_{\text{bin}}^{3/2}} \tag{35}$$

$$= 103 \, \text{yr} \left( \frac{m_{\text{bin}}}{30 \, M_\odot} \right)^{-3/2} \left( \frac{a_1}{0.1 \, \text{AU}} \right)^{5/2} \left( 1 - e_1^2 \right).$$

If the precession timescale of the inner binary is shorter than the quadrupole timescale, eccentricity excitations can be suppressed (e.g., [17,20,352,380,381]). If the GR timescale is comparable to the octupole timescale, the system can be excited to larger eccentricities than

the ones reached without the GR effects (e.g., [20,382]. A similar result takes place for the hexadecapole level of approximation $n = 4$, in Equation (32) (e.g., [383]).

**Spin effects, 1.5pN**

The compact objects' spins can cause a variety of precessions that may affect the overall dynamics. For example, de-Sitter precession (e.g., [384]) can cause the precession of the compact object spin vector about the angular momentum of the binary. On the other hand, Lense–Thirring precession can cause the precession of the angular momentum about the spin of the compact object (e.g., [384]), if the two vectors are initially misaligned. This effect translates into eccentricity exaction as the angular momentum changes.

We will see in Section 8 that these effects may leave an imprint detectable by future GW detectors, such as LISA.

During the EKL mechanism, the binary's spin axes undergo chaotic evolution, similar to the non-compact object case (e.g., [335,337,340,385,386]). This process leads to a wide range of the final spin–orbit misalignment ($0°$–$180°$), thus favouring the formation of merging binaries with misaligned spins.

Unlike strong scatterings, which dominate mergers in massive clusters and produce an isotropic distribution of spin orientations, the compact object spins in this EKL+1.5pN case are strongly correlated with one another (e.g., [387]).

**GW, 2.5pN**

The higher pN terms affect the gravitational wave (GW) emission of compact object binaries (e.g., [388]). The large eccentricity, potentially produced via the octupole level of approximation, can produce pulse-like GW emissions, which shrink the binary semimajor axis and later circularise the orbit (e.g., [389,390]).

**Resonant relaxation processes**

Binaries embedded inside a dense stellar cluster are subjected to a continuous influence from the gravitational field generated by all the other cluster members, which can impinge secular effects on their orbital evolution (e.g., [203,391–394]). These effects include different relaxation processes and the precession of the orbit due to the extended and possibly anisotropic gravitational potential.

Within the sphere of influence of the SMBH, the motion of stars and compact objects is nearly Keplerian; therefore, the encounters between stars are correlated. These correlated gravitational encounters result in torques, which can change both the direction and magnitude of the angular momentum of the orbit of the binary about the SMBH, known as resonant relaxation processes (e.g., [391,393,395–400]).

*Resonant relaxation*

This process causes the variation of the magnitude and direction of the outer angular momentum (thus, the eccentricity) over a timescale (e.g., [391,393,401])

$$t_{\text{rr,s}} = 0.9 \text{Gyr} \left( \frac{M_{\text{SMBH}}}{4.3 \times 10^6 \text{ M}_\odot} \right)^{1/2} \left( \frac{a_2}{0.1 \text{ pc}} \right)^{3/2} \left( \frac{m}{1 \text{ M}_\odot} \right)^{-1} \tag{36}$$

*Vector resonant relaxation*

This process tends to alter the orientation of the angular momentum, thus changing the binary's mutual inclination between the inner and the outer orbit over a timescale (e.g., [391,393,402])

$$t_{\text{rr,v}} = N^{-1/2} t_{\text{rr,s}}, \tag{37}$$

where $N$ is the number of stars within the outer orbit $a_2$. While the vector resonant relaxation rate depends on the underlying density distribution profile, it may be comparable to the EKL timescale at distances of $\leq 0.5$ pc, depending on the binary's separation [402]. Overall, vector resonant relaxation may work to enhance the merger rate by changing

mutual inclination to a more EKL-favoured regime of the parameter space [402]. Given the uncertainties, this regime yields a BH merger rate similar to the one expected from the inner $\leq 0.1$ pc region.

For even further distances from the SMBH, at the edge of the sphere of influence, the deviation from the spherical potential of the nuclear star cluster can result in a long-term effect, similar to the EKL [394]. This processes also yields a similar BH merger rate as the one expected from the inner $\leq 0.1$ pc merger rate. Below, we discuss the overall expected merger rate and the relevant uncertainties.

### 6.3. Initial Conditions and Unknowns

The description of the interplay among COs, COBs, and a central SMBH relies upon several unknowns, which may significantly affect the possible development of both COB and SMBH–CO mergers.

The number of binaries, their period and eccentricity distribution, and the CO mass distribution are poorly constrained by observations and largely influence the final outcomes of the dynamics. The density distribution of COs around the SMBH is another crucial, but poorly constrained, quantity that affects all dynamical processes, from relaxation to binary hardening (e.g., [48]), dynamical friction (e.g., [269]), to the collision rate (e.g., [51,311]).

Since binary's inner and outer period distribution are degenerate, the EKL efficiency is less sensitive to the aforementioned uncertainties. While at first glance, this may be counterintuitive, it can be understood when considering the stability requirement. Long-term stability yields a hierarchical configuration and avoids the breakup of the binary due to the SMBH's tidal forces (i.e., the Hills mechanism [284]). Therefore, binaries closer to the SMBH may have a shorter inner period than those further away from the SMBH (e.g., [205]).

Regarding the formation of COB mergers, surely one important parameter is represented by the number of binaries that can actually form (see [292]), which, in turn, depends on the star formation rate in the galactic centre and the rate at which infalling clusters can supply new COs and COBs to the nucleus [278].

The Milky Way's centre represents our benchmark and, in principle, can be used to estimate the star formation rate inside an NC. Within the SMBH's sphere of influence, the Milky Way contains about $\sim 10^7$ M$_\odot$ stars and stellar remnants (e.g., [403,404]), assuming a continuous star formation rate and an age of the galaxy of about 10 Gyr; this gives an approximate rate of $10^3$ M$_\odot$ Myr$^{-1}$. Despite this rough estimate being consistent with recent observations and theoretical advancements (e.g., [204,206,405]), it remains highly uncertain and should be taken with a grain of salt.

## 7. Dynamics of Black Hole Binaries in Active Galactic Nuclei

The presence of an accretion disc feeding a central SMBH is a natural requirement to explain the luminosity, variability, and spectra of observed AGNs, although both the pathways that can lead to the accretion disc's formation and the physics regulating its evolution are still not fully understood. In the classical framework depicted by Shakura and Sunyaev [406] and extended by Novikov and Thorne [407] to include general relativistic effects, the disc settles into a stationary configuration owing to inwards angular momentum transport due to an effective viscosity in the disc.

Above a critical value of the disc's surface density, the angular momentum and energy transfer that a star experiences while passing through the disc become significant, causing dissipation, which can *capture* the star, forcing its orbit to settle into the disc plane and undergo inward migration, causing a steepening of the radial density profile [408–411]. Two-body relaxation with objects outside the disc competes with this dissipative process, leading to an almost stationary state where the two-body relaxation scatters stellar objects off the disc, whilst dissipation captures them [412]. When the timescales of these competing process are comparable, solar-mass objects are more easily accreted onto the central SMBH, possibly enhancing the rate of tidal disruption events [27,413] and naturally leading to the

formation of a nuclear stellar disc [29]. Once stars and stellar compact objects are captured into the disc, the relative velocity among the disc–objects decreases, leading to a significant increase of the collision rate, $\sigma_c \propto v_{\rm rel}^{-2}$, especially in the outer regions of the disc, possibly favouring the formation of an IMBH seed (e.g., [26]).

Depending on the disc's properties, the swift growth of an IMBH seed via gas accretion can be so efficient to deplete gas from the AGN's disc, a feature that is inconsistent with the properties of observed AGNs. However, winds and jets launched by the accreting IMBH could create a hole around the IMBH. This feedback mechanism could quench the IMBH's growth in MW-sized galaxies and almost eradicate the disc in AGNs with an SMBH lighter than $10^5$ M$_\odot$, thus explaining the dearth of high-Eddington ratio AGNs in that SMBH mass range [414].

The stellar object motion is driven by different forces owed to accretion, dynamical friction exerted by the gaseous medium, and stellar scatterings [33,415].

Whilst moving in the disc, the BH motion is subject to migration induced by the resonant gravitational interaction with the gas disc [416]. If the gravitational torque of the moving body is smaller than the gas viscous torque, the body undergoes the so-called Type I migration, triggered by Lindblad and corotation resonances [25,416–418]. The typical timescale of Type I migration depends on the mass of the object and the central BH, the AGN surface density and thickness, and the location of the captured object.

Sufficiently large gravitational torques, instead, enable the moving object to open a gap in the disc. In such a case, the object migrates due to the torque exerted by the gas on the gap boundary, leading to the Type II migration [417,419–421]. The timescale of Type II migration is generally larger than that of Type I migration, depending on the SMBH's mass, the moving object, and the viscosity of the gas. Non-axisymmetric streamers flowing across the gap may significantly alter the migration rate [421,422].

Depending on the gas structure and properties, gas torques can also push the object away from the disc centre, thus causing an outward migration [423,424]. Inward and outward torques can cancel out in particular regions of the disc and form a so-called *migration trap*, which halts the migration. Migration traps are natural predictions of protoplanetary disc models [425] and could be a possible mechanism to form the cores of giant gaseous planets [426]. The location of the trap inside the disc may be independent of the SMBH's mass and the SMBH-to-BH mass ratio, as these quantities only affect the amplitude of the torque [86]. However, the details of migration trap formation are still unclear and are mostly connected with the study of protoplanetary disc formation: inefficient viscous mixing can cause torque saturation, damping migration for sufficiently small objects [427] and leading to a mass dependency on the location of the traps [428–430]; the establishment of a dynamic torque can halt (boost) inward(outward) migration [431,432], whilst a heating torque derived from accretion processes onto the moving body can counteract inward migration [433–435]. Whilst some elements of protoplanetary disc dynamics can be borrowed to explain compact objects' dynamics in AGNs, any parallelism must be cautionary, taking into account that the latter are generally comprised of hotter, high-viscosity, turbulent gas, where torque saturation is less likely [436,437].

Depending on the disc model adopted, these migration traps are set at tens- and a few hundred-times the SMBH gravitational radius, i.e., $\sim$ (0.4–3) mpc for an SMBH mass $M_{\rm SMBH} = 10^8$ M$_\odot$ [86].

More generally, even if such migration traps do not exist in AGN discs [436,437], a similar accumulation of objects may take place at radii where the inward migration rate becomes slower than the rate at which objects are captured in the disc. This may happen in regions where the physical properties of the disc change (e.g., source of opacity, self-gravity, etc.) or where gaps are opened by the gravitational torques of objects in the disc [33]. Binary formation and mergers may be especially abundant in these regions.

Once BHs and COs have settled in the disc, the formation of binaries can be triggered by three- and few-body interactions as in galaxies harbouring a quiescent SMBH and by single–single interactions triggered by gas dissipation [33]. The three-body scattering

process can efficiently aid binary formation in AGN discs, owing to the fact that gas damps stellar velocities via gaseous dynamical friction and torques [33], thus providing low velocities and high densities. Further, the presence of a gaseous medium offers a further mechanism: gas capture binary formation. In such a mechanism, two objects passing through their mutual Hill spheres can bind together if the Hill sphere crossing time is longer than the timescale over which gas dynamical friction damps the relative velocities of the two objects [438]. As discussed in [33], the gas-assisted binary formation rate steeply increases toward the disc's inner regions and clearly overtakes the formation rate associated with three-body scattering. Thus, gas capture binary formation is expected to be the dominant process for binary formation in AGN discs. This also favours the mass growth of IMBH-sized objects via gas accretion and repeated mergers with stars and compact objects in the disc [26,33,34,436,439].

All these processes are nicely sketched in Figure 13, a schematic view on binary formation activities in AGNs described in [33].

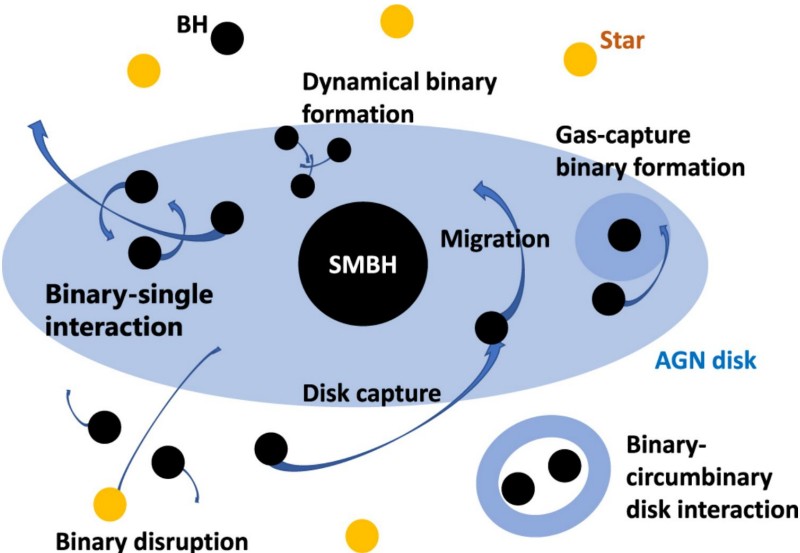

**Figure 13.** Schematic illustration of the different processes that contribute to black hole binary formation in the disc of an active galactic nucleus. Taken from Tagawa et al, ApJ, 2020, 898(1), 25 ©AAS. [33]. Reproduced with permission.

Once binaries start forming in the disc, their long-term evolution is regulated by both dynamical and gaseous processes. Having a relatively large cross-section compared to single objects, newborn hard binaries will further harden via binary–single scattering [11,210,283,329]. Binary–single interactions with stars or compact objects in the spherical star cluster typically kick the binary from the gaseous disc to inclined orbits, while gas dynamical friction drives the binaries to settle back to the disc. Furthermore, binary–single interactions increase the binary eccentricity in a disc configuration more than in isotropic scatterings, which accelerates the subsequent GW-driven merger [35,440], and are expected to work even more efficiently close to migration traps, where the binary–single interaction rate increases significantly [436,441]. Aside from binary–single interactions, which represent the dominant process in binary hardening [30,33], the gaseous medium in which the binary is embedded can trigger further binary hardening owing to dynamical friction [28] and Type I/II torques exerted from a circumbinary disc formed around the moving binary [25,420,441–443].

Despite the many processes likely at play in an AGN, one seems clearly dominating in determining BBH formation, namely the gas capture process, which accounts for 64–85% of all BBH mergers occurring around an MW-like SMBH [33]. Once formed, binaries merge very efficiently in AGNs due to binary–single interactions, even if gas-driven migration effects and accretion effects are completely neglected [33].

## 8. Black Hole and Neutron Star Mergers around Supermassive Black Holes: Implications for Current and Future Gravitational Wave Detections

The various physical processes described in the previous sections leave an imprint on the population of merging COs. In the following, we discuss in more detail what properties characterise the population of merging BBHs and NS–BH binaries formed in galactic nuclei, differentiating between quiescent SMBHs and AGNs, as well as the consequences for current and future GW detectors.

In these regards, it is worth recalling here that eccentric COBs emit broadband GWs with a characteristic peak frequency, which can be expressed as [163,444]

$$f_{\text{peak}} = 4.6 \, \text{Hz} \left( \frac{k}{30} \right)^{-3/2} \left( \frac{m_{\text{bin}}}{30 \, \text{M}_\odot} \right)^{-1} \zeta(e_{\text{bin}}),$$ (38)

where $k = a_{\text{bin}}/R_{\text{Schw}} = c^2 a_{\text{bin}}/(2Gm_{\text{bin}})$ represents the ratio between the binary semimajor axis and the COB total Schwarzschild radius, in such a way that $k = 3$ corresponds to the innermost stable circular orbit (ISCO) for non-spinning BHs. There are different expressions for the function $\zeta(e_{\text{bin}})$ [264,444]:

$$\zeta(e_{\text{bin}}) = \begin{cases} (1 + e_{\text{bin}})^{1.1954}/(1 - e_{\text{bin}}^2)^{3/2} \\ (1 + e_{\text{bin}})^{1/2}/(1 - e_{\text{bin}})^{3/2} \end{cases}$$ (39)

From the equation above, we see that a typical hard BBH with mass $m_{\text{bin}} = 30 \, \text{M}_\odot$ and $a_{\text{hard}} = 0.1$ AU, i.e., $k = 9 \times 10^6$, will emit GWs with the frequency peaking at $f \sim 0.35$ mHz.

Thus, merging BBHs represent potentially multiband GW sources (e.g., [445]), which could be visible to both low-frequency GW detectors such as LISA [446–448] and TianQin [449], Taiji [450,451], mid-range frequencies such as ALIA [452], DECIGO [453,454], and other decihertz observatories [455–457], high-frequency detectors such as LIGO-Virgo-KAGRA [458,459], and future detectors such as the Einstein Telescope [460,461] or Cosmic Explorer [462,463].

The next section will focus on the properties of COB mergers in galactic nuclei, attempting to highlight how they could be used to identify the COB origin in different GW detectors.

### 8.1. Population Properties: Masses, Mass Ratio, Spins, and Eccentricity

There are several parameters that could help in unravelling BBH mergers with a dynamical origin, although the uncertainties in the theoretical models for both stellar evolution and the dynamics make it harder to find a truly unique spot where dynamical mergers stand apart from the isolated ones.

Merging COBs can be characterised primarily through intrinsic quantities, such as the $j$-th component mass $m_j$ and adimensional spin $\chi_j = \frac{cS_j}{Gm_j^2}$ (with $S_j$ the spin), as well as the binary eccentricity $e$ and semimajor axis $a$. Nonetheless, there are further quantities that are particularly important from the perspective of GW detection. Among others, the so-called chirp mass, a parameter that, at the lowest order, can be linked to the frequency ($f$) and its variation ($\dot{f}$) of the emitted GW via the relation:

$$\mathcal{M} = (m_1 m_2)^{3/5}/(m_1 + m_2)^{1/5} = \frac{c^3}{G} \left( \frac{5}{96} \pi^{-8/3} f^{-11/3} \dot{f} \right)^{3/5},$$ (40)

thus can be directly measured by GW detectors [464–466].

Similarly, also spins affect the evolution in the phase and amplitude of the emitted GWs, but it is hard to fully untangle the two spin vectors [467,468], even in space-borne detectors such as LISA (see, e.g., [465,466]). Nonetheless, it is still possible to retrieve information encoded in the dominant spin effect by using a one-dimensional parametrisation,

which, in its simplest form, can be written as a simple mass-weighted linear projection of the spins onto the binary orbital momentum vector $\vec{L}$ [469–473]:

$$\chi_{\text{eff}} = \frac{(m_1\vec{\chi}_1 + m_2\vec{\chi}_2) \cdot \tilde{\mathbf{L}}_{\text{bin}}}{m_1 + m_2} = \frac{m_1\chi_1 L \cos(\theta_1) + m_2\chi_2 L \cos(\theta_2)}{m_{\text{bin}}}, \tag{41}$$

with $\theta_j$ the angle between the binary angular momentum and the spin of the $j$-th binary component. As suggested by the BBH mergers detected during the three LVC observation runs [141], this parameter seems to cluster around a low value (e.g., [474–478]).

Spins' information is encoded also in another measurable quantity, unfortunately poorly constrained in current detectors [68]: the precession spin parameter $\chi_p$ [479], which measures the degree of in-plane spin and parametrises the rate of the relativistic precession of the binary orbital plane [141,480]:

$$\chi_p = \max\left[\chi_1 \sin\theta_1, \left(\frac{3+4q}{4+3q}\right) q\chi_2 \sin\theta_2\right]. \tag{42}$$

Aside from masses and spins, another parameter that could be measured from GW detectors is the binary eccentricity. This became possible only relatively recently (e.g., [143,144,481]), because binary parameter estimation from the detected signal exploits Bayesian inference, which can be extremely computationally demanding. This led to the development of, on the one hand, numerical relativity simulations aimed at obtaining waveform templates [150] and, on the other hand, approximate methods, or *approximants*, which greatly reduce the computational cost of the data analysis, but generally are constructed on the leading order of the GW signal expansion (e.g., [481]). A recently developed technique, however, enables the possibility to retrieve information on the binary eccentricity at a small computational cost [482,483] and permits placing constraints on the eccentricity of LVC sources, suggesting that up to four of them might be eccentric sources [146,147]. In the frequency range of LVC detectors, i.e., $f \sim 10$ Hz, it has been shown that, at design sensitivity, it would be possible to discern an eccentric source provided that $e_{10\text{Hz}} > 0.04$, the error on the eccentricity measurement being around $\delta e = \sim (10^{-4} - 10^{-3})(D/100\text{Mpc})$ [151,152].

Merging COBs developing in different environments may exhibit peculiar values of the aforementioned quantities; thus, their measurement is crucial to assess the origin of GW sources. For example, mergers with one or both components in the upper mass-gap, such as GW190521 [484], could represent the best candidates to identify signatures of a dynamical origin.

Generally, the formation of mass-gap BH mergers in the isolated binary scenario is rather unlikely, whilst in dynamical environments, hierarchical mergers represent a viable possibility to explain them, especially in the case of galactic nuclei. Close to an SMBH, the escape velocity is roughly given by

$$v_{\text{esc}} = 2076\text{km s}^{-1}\left(\frac{M_{\text{SMBH}}}{10^6\ \text{M}_\odot}\right)^{1/2}\left(\frac{r}{1\text{mpc}}\right)^{-1/2}; \tag{43}$$

thus, close to an MW-sized SMBH, the escape velocity can favour the retention of first-generation merger products (1 g) or second-generation BHs (2g), which are subjected to GW recoil kicks as large as $10^3$ km s$^{-1}$. Once recoiled, the BH will eventually come back into the galactic centre over a mass segregation time, form a new binary via binary capture [13] or three-body scattering [310], harden, and merge. Depending on the nucleus properties, the whole process can take much longer than the Hubble time, owing to the steep dependence of the dynamical timescales (Equations (4), (25), and 28) on the nucleus velocity dispersion and density. Nonetheless, the development of mergers involving 2 g or 1 g BHs is more likely in galactic nuclei—O(10%)—than in globular or young massive clusters [98,99,111,116,127,485,486].

To discuss other potential characteristic traits of COB mergers in galactic nuclei, let us divide the processes that can trigger a merger into three main categories: (i) gravitational scatterings, (ii) EKL-driven, and (iii) AGN-driven.

### 8.1.1. Gravitational Scatterings

Gravitational scatterings are likely the main engine triggering COB mergers around SMBHs in the $M_{\rm SMBH} = (10^6 - 10^8)\,{\rm M}_\odot$ mass range, being responsible for up to 20–70% of mergers in galactic nuclei [30,278,440,487]. BBH merger products formed via binary–single scatterings affect the overall BH mass spectrum, populating the upper mass-gap and extending beyond $M_{\rm BH} \gtrsim 100{\rm M}_\odot$ [114,116,278,448,485]. Binary–single scatterings in galactic nuclei can also favour the formation of NS–BH binary mergers, the individual merger rate—i.e., the number of mergers per cluster—being around 10(100)-times larger compared to globular (young) clusters (e.g., [130,136,311]). A substantial fraction of NS–BH mergers forming in galactic nuclei, and in dense star clusters in general, is expected to clearly differ from those formed in isolation, having larger chirp masses and primary components with a mass $m_1 > 20\,{\rm M}_\odot$ [130,132]. Binary–single scatterings and GW captures are efficient mechanisms to form eccentric binaries: $50 - 90\%$ BBH mergers formed this way have an eccentricity $e_{10{\rm Hz}} > 0.9$ when sweeping through the typical LIGO band [151,152,264,488].

The geometry of the scattering can be rather important in determining the merger characteristics: if the scattering occurs in a plane, as can happen in AGN discs, for example, the merger rate is boosted by a factor 10–100 owing to the fact that the merger efficiency in three-body scattering increases at decreasing binary–single object mutual inclination [35,440]. The enhancement in the merger rate is also accompanied by an enhancement in the probability to form merging binaries with a high eccentricity in the LVC band [35]. Mergers induced by a "single" binary–single scattering, i.e., in which the merger occurs before the next interaction takes place, are more likely to emit GWs in the 1-10 mHz frequency range, where LISA and space-borne detectors are sensitive [35], whilst those formed chaotically during a three-body resonant interaction, or scramble, and via GW captures typically quickly merge in the 1-10 Hz band, where LIGO and ground-based detectors are sensitive [35,264]. Given the chaotic nature of the gravitational scattering process, mergers formed this way are expected to feature an isotropic distribution of their spin orientation, which implies generally small $\chi_{\rm eff}$ values. For comparison, fully (anti)aligned spins implies (negative) positive values of $\chi_{\rm eff}$.

### 8.1.2. Eccentric Kozai–Lidov Mechanism

When the effect of the SMBH tidal field cannot be neglected, the EKL mechanism can aid the COB coalescence through eccentricity excitation. The merger efficiency for the EKL-driven channel weakly increases with the SMBH mass in the $M_{\rm SMBHs} = (10^6 - 10^8)\,{\rm M}_\odot$ mass range and decreases for heavier SMBH mass values [101,278,292,401,487].

Around 40% of EKL-assisted mergers have primary masses in the upper mass-gap, $m_1 = (50 - 90)\,{\rm M}_\odot$, and around 30% have also a companion with mass $m_2 < 20\,{\rm M}_\odot$, thus mass ratios $q \simeq (0.1$–$0.6)$ [88,278]. Only a fraction $\sim 1 - 20\%$ of EKL-driven mergers are expected to preserve an eccentricity $e > 0.1$ at $> 10Hz$ [379,401], owing to the fact that the eccentricity increase driven by the EKL mechanism can drive the merger of binaries that are initially relatively loose. Nonetheless, around 40% of these binaries may appear eccentric in the LISA band, and up to 5% of them could preserve an $e > 0.1$ while sweeping through the deci-Hz band, thus representing potentially bright multiband sources [98]. Although LISA is unlikely to observe more than a few tens of BBH mergers (e.g., [359,489]), it could be possible to observe eccentricity variation for BBHs in the Galactic Centre [490].

The development of EKL resonances and the consequent increase of the binary eccentricity up to a maximum value $e_{\rm max}$ reduces the merging timescale, possibly affecting the overall delay time of EKL-driven mergers. In such a case, the merger time can be expressed as [379,491,492]

$$t_{\rm GW} = t_{\rm GW,0}(1 - e_{\rm max}^2)^\alpha, \tag{44}$$

with $\alpha = 1.5 - 2 - 2.5 - 3$ in the eccentricity ranges $e_{max} = (0 - 0.6)$, $(0.6 - 0.8)$, $(0.8 - 0.95)$, and $(0.95 - 1)$, respectively [379,491,492]. This implies that the larger the maximum eccentricity, the shorter the merging time is. Let us consider an initially circular, equal-mass binary with $m_{bin} = 30$ M$_\odot$ and semimajor axis $a = 0.1$ AU: its merging time is $t_{GW,c} = 4.8$ Gyr [388]. Eccentricity increases owing to EKL up to $e_{max} = 0.5(0.9)$; thus, the binary merges after $t_{GW} = 1.7(0.02)$ Gyr, a notable difference that highlights how crucial the orbital distribution of compact objects around an SMBH is.

If the SMBH is spinning, as indicated by several observations (e.g., [493,494]), the coupling among the SMBH spin and the BBH–SMBH and BBH angular momenta can cause an efficient *apsidal precession resonance*, meaning that the outer and inner binaries have the same pericentre precession rate owing to both Newtonian and general relativistic effects [495]. This results in an angular momentum exchange between the BBH–SMBH and the BBH, which leads to a significant growth of the BBH eccentricity up to a maximum value that increases at increasing the BBH–SMBH initial eccentricity [495–498]. This mechanism can lead to an increase of the BBH even in cases in which the EKL resonance does not develop, e.g., nearly coplanar systems, and could leave an imprint in the merging BBH waveform, which might be detectable with LISA and help constrain the SMBH's spin amplitude and its mass [496,499,500].

A binary undergoing EKL around a Kerr SMBH can undergo de-Sitter and Lense–Thirring precession mechanisms, which cause the chaotic re-orientation of the spins (e.g., [387]), leading to merging binaries with misaligned spins and nearly zero $\chi_{eff}$ (e.g., [96,98,99,387,491,492,501,502]).

### 8.1.3. Active Galactic Nuclei

Merging BBHs in AGN discs are expected to exhibit several peculiarities, compared to those developing in quiescent nuclei via EKL oscillations or gravitational scatterings.

Given the large escape velocities in AGNs, repeated mergers can be quite common in such environments, thus providing a further channel to produce mass-gap objects. If migration traps do exist, repeated mergers might constitute up to 20–50% of all COB mergers [32,503], with more than 40% of all mergers involving one BH with mass $\geq 50$ M$_\odot$ [32]. Even in the absence of migration traps, hierarchical mergers can constitute 20–45% of all mergers mediated by binary–single interactions and gaseous torques [33]. Nonetheless, the fraction of remnants in the high end of the BH mass distribution is expected to remain generally low, around 3–11%, but can significantly vary depending on the adopted model [34]. The high occurrence of repeated mergers influences the mass ratio of AGN mergers, whose distribution peaks at values around $q = 0.2 - 0.7$, depending on the models [31,503]. Similar to quiescent nuclei, dynamical interactions can enable the formation of eccentric mergers also in AGNs via either binary–single interactions or GW captures. Given the flattened configuration of AGN discs, most of the mergers produced via binary–single scatterings are expected to produce eccentric binaries detectable by LISA [440]. Together with binary–single interactions, GW captures occurring within $O(mpc)$ from the SMBH can be sufficiently energetic to trigger the formation of mergers preserving an eccentricity $e > 0.3$ in the LIGO-Virgo band with a probability of 20–70% [35,440].

Gaseous torques and accretion are expected to align both the BH spin vectors and the binary angular momentum, thus suggesting that BBHs in AGNs are characterised by either aligned or antialigned spins. BBHs moving on prograde orbits with respect to the AGN angular momentum are expected to rapidly accrete gas, spin up, and align their spins over a timescale shorter than the AGN's lifetime [504], whilst those on retrograde orbits are more likely to merge and leave behind a final BH with a spin antialigned with respect to the disc [25]. Given the alignment/antialignment of the orbital spins, AGN mergers are expected to have an effective spin distribution peaked at positive/negative values, likely around 0.4 [32]. The preponderance of hierarchical mergers would also favour the development of antialigned/aligned mergers [32], thus possibly causing the formation of

a sub-population of mergers characterised by large masses and $\chi_{\rm eff} < 0$, a feature hardly reproducible through other channels. Nonetheless, even in AGNs, it is possible to generate low-$\chi_{\rm eff}$ mergers, depending on the efficiency of radial migration and the binary–single scattering rate [505], although the overall distribution could preserve a small excess toward positive values dominated by mergers among high-generation BHs [503]. Gaseous accretion can also favour the formation of COs in the lower mass-gap, thus providing a suitable explanation for GW sources such as GW190425 or GW190814 [34,134].

As shown in some numerical models, in AGN mergers, a combination of the effective spin parameter and the precession parameter, namely $\chi_{\rm tot} \equiv (\chi_{\rm eff}^2 + \chi_p^2)^{1/2}$, increases with the merger chirp mass up to the maximum chirp mass allowed by the adopted, first-generation, mass spectrum and then saturates toward a value $\chi_{\rm typ} \sim 0.6$ [506]. Despite that such a typical quantity could represent an optimal parameter to identify AGN merger candidates, the precession spin parameter $\chi_p$ is often unconstrained in observed data, making it hard to place reliable constraints on $\chi_{\rm typ}$.

One of the most-intriguing predictions of the AGN channel is the possible production of EM signatures associated with the merger event. A well-known example is the GW190521 source, which has been proposed to be associated with an EM transient detected by the Zwicky Transient Facility [507], although it is rather hard to assess whether or not the GW and EM signals are truly associated [508,509], as GW190521 has a localisation volume that likely contains $\sim 10^4$ AGNs [510]. Nonetheless, future follow-up campaigns targeting the most-massive BBH mergers could establish their association with an EM counterpart, provided that the fraction of BBHs producing an AGN flare is substantial (>0.1) [510].

Establishing whether a BBH merger has an AGN origin could be assessed statistically on the basis of the spatial correlation, a technique that requires at least $f_{\rm AGN} N_{\rm det}$ detections, where $f_{\rm AGN}$ corresponds to the fraction of GWs coming from AGNs [511]. In principle, this implies that the missing evidence of a BBH–AGN connection in the light of 100 GW detections would suggest $f_{\rm AGN} < 0.25$, although the number of required detections to assess their AGN origin hugely varies with the AGN number density [511].

*8.2. Expected Merger Rate and Prediction*

In the following, we retrieved from the literature merger rate estimates at redshift $z = 0$ for both BBH and NS–BH mergers induced by dynamics around a quiescent SMBH or in an AGN. When the authors provided the number of events per galaxy and per year (e.g., [264]), we multiplied this quantity by the local density of Milky Way-equivalent galaxies $\rho_{\rm MWEG} = 0.0116$ Mpc$^{-3}$ [512,513]. Figures 14 and 15 show the merger rate density interval for BBH and NS–BH mergers at redshift $z \sim 0$ reported in the literature.

Broadly speaking, the inferred merger rates for mergers around quiescent SMBHs and AGNs cover a wide range of values, up to 3–4 orders of magnitude, owing to the large uncertainties affecting the models.

For AGNs, there are several sources of uncertainties (for a detailed discussion, see [33,514]): the number density of galactic nuclei hosting an AGN, $(10^{-3} - 10^{-2})$ Mpc$^{-3}$ (e.g., [515]), the BH number density in galactic nuclei $(10^{4-6})$ pc$^{-3}$ [53,55,58], the AGN occupation fraction (0.01–0.3) [516], the BH binary fraction (0.01–0.2) [55,288,436,517], and the AGN lifetime (0.1–100) Myr [518–520].

For quiescent SMBHs, instead, the uncertainties are owed mostly to the number of BHs lurking in the galactic centre and the rate at which the BH population is replenished [278,292,401], the geometry of the NC surrounding the SMBH [292,394], the SMBH's mass [278,292,401], and the properties of the BBH population [278,487].

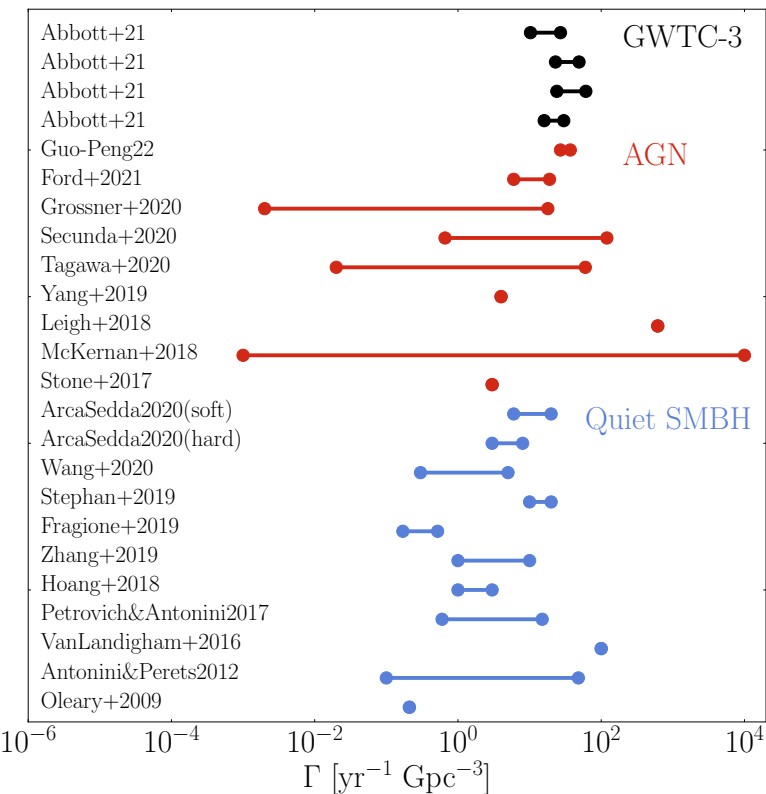

**Figure 14.** Merger rate density of BH–BH mergers in quiescent and active galactic nuclei at redshift $z = 0$.

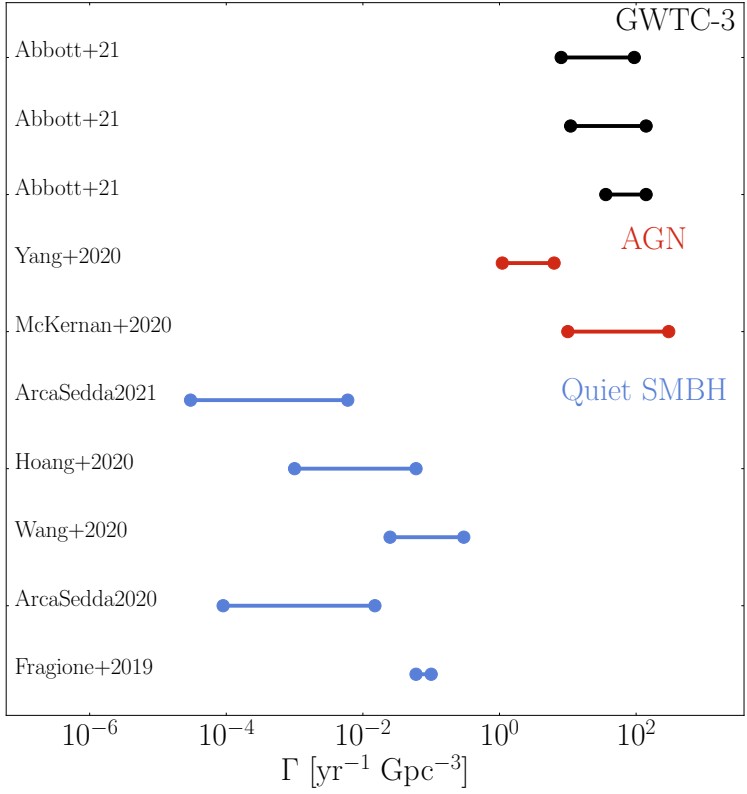

**Figure 15.** Same as in Figure 14, but for NS–BH mergers.

Adopting Milky Way-like conditions and assuming that the fraction of galaxies hosting an NC is ~0.5 yield a BBH merger rate of $\sim 1$–$20$ Gpc$^{-3}$ yr$^{-1}$ (e.g., [59,359]). This rate is consistent with a simplified model that starts with objects at the BH stage (e.g., [292] and N-body calculations [278]. Vector resonant relaxation only slightly enhance the rate, for Milky Way-like conditions, but is much more effective for a smaller-mass SMBH (e.g., [402]). On the other had, the EKL derivative for a non-spherical potential yields a rate of $\sim 1$–$15$ Gpc$^{-3}$ yr$^{-1}$ [394]. Thus, combining these three effects, the merger rate from within the inner few to ten parsecs at the centre of a galaxy is expected to be $1$–$50$ Gpc$^{-3}$ yr$^{-1}$. The lower limit in this range is taken to be from the lower limit of one of the channels, and the upper limit is taken from the combination of the three chancels.

Future detectors, such as LISA or DECIGO, could shed light on the population of merging binaries in our Galactic Centre. For example, recent models of the Milky Way's centre suggest that LISA could identify up to $1 - 20(15 - 150)$ BBH (WD-WD binaries) [359].

Thus, the merger rate boundary limits for BBHs around quiescent SMBHs, $\mathcal{R}(0) \simeq (0.1 - 10^2)$yr$^{-1}$Gpc$^{-3}$, is somewhat narrower with respect to AGNs—whose rate spans up to seven orders of magnitudes in some models. Future numerical simulations that account simultaneously for the evolution of the galactic nucleus and its stellar population could help to further reduce the uncertainties.

Another interesting class of GW sources is represented by NS–BH mergers, which may also produce EM emission promptly after the merger. However, the formation of NS–BH binaries is strongly suppressed in dense stellar environments, owing to the fact that BHs have larger cross-sections than other COs, thus favouring the interaction among themselves. This implies that the formation of NS–BH binaries either takes place on long timescales or is a by-product of primordial binaries. Either way, merging NS–BH binaries in quiescent galactic nuclei can form via binary–single scatterings, with typical merger rates of $\mathcal{R} \sim 0.0001$–$0.01$yr$^{-1}$Gpc$^{-3}$ [130,136], similar to what happens in globular and young clusters [132,133]. Merging NS–BH forming in galactic nuclei possibly contribute to the formation of low-mass ratio systems such as GW190814 [136].

Mergers can be triggered by EKL oscillations induced onto binaries formed via GW captures ($\mathcal{R} \sim (0.001$–$0.06)$ yr$^{-1}$Gpc$^{-3}$) [311], dynamical interactions ($\mathcal{R} \sim 0.06$–$0.1$ yr$^{-1}$Gpc$^{-3}$) [401], or binary stellar evolution—although likely with a rather low ($\sim 0.6\%$) probability ($\mathcal{R} \sim 0.2$–$2$ yr$^{-1}$Gpc$^{-3}$ [59,359].

In AGN discs, instead, the production of NS–BH mergers proceeds, in principle, similarly to BBHs, but their amount relative to BBHs is rather uncertain. In general, double-NS (DNS) and NS–BH mergers constitute a relatively small fraction among all mergers developing in an AGN, namely $f_{\mathrm{DNS}} \simeq (0.5 - 3) \times 10^{-4}$ (DNS) and $f_{\mathrm{NSBH}} = (0.8 - 6) \times 10^{-4}$ (NS–BH) [34], although, according to some models, these quantities can be as large as $f_{\mathrm{DNS}} = 0.01(f_{\mathrm{NSBH}} = 0.3)$ for DNS (NS–BH) mergers [521]. These estimates correspond to a merger rate density of $f_{\mathrm{DNS}} = 400 f_{\mathrm{AGN}}$yr$^{-1}$Gpc$^{-3}$ (DNS) and $f_{\mathrm{NS-BH}} = 10 - 300 f_{\mathrm{AGN}}$yr$^{-1}$Gpc$^{-3}$ (NS–BH) [521], although, depending on the model assumptions, these estimates can be up to 2–3 orders of magnitude smaller [34].

A list of the values for the BBH and NS–BH merger rate density is reported in Tables 1 and 2.

Comparing the modelled merger rate densities listed in Tables 1 and 2 with that inferred from GW observations [141] makes evident that the galactic nuclei channel for BBH mergers can contribute quite significantly to the overall population, but is unlikely to contribute much to the global population of NS–BH mergers.

**Table 1.** BBH merger rate density for quiescent SMBHs and AGNs. Column 1: type of physical processes included—Kozai–Lidov (EKL), dynamical scatterings (single–single or binary–single, DYN), binary stellar evolution (SEV), gaseous capture via migration and/or dynamical friction (GAS), migration (MIG), and pairing in migration traps (TRP). Column 2: merger rate density. Column 3: reference. When a volumetric rate density was not provided in the referenced paper, we adopted, when required, a local galaxy number density of $\rho_{glx} = 0.0116$ Mpc$^{-3}$ [512,513] and an average number of BBH binaries in an MW-like nucleus of $N_{BBH} = 200$ [278,292].

| BBH Merger Rates | | |
|---|---|---|
| **Process** | **$\Gamma$ (yr$^{-1}$Gpc$^{-3}$)** | **Ref.** |
| GWTC-3 | | |
| - | **17.9–44** | **The LIGO Scientific Collaboration et al. [141]** |
| Quiescent SMBH | | |
| EKL+SEV | 4–24 | Wang et al. [359] |
| EKL+DYN | 3–8 | Arca Sedda [278] |
| EKL+DYN | 6–20 | Arca Sedda [278] |
| EKL+SEV | 10–20 | Stephan et al. [59] |
| EKL | 0.17–0.52 | Fragione et al. [401] |
| EKL+DYN | 1–10 | Zhang et al. [487] |
| EKL | 1–3 | Hoang et al. [292] |
| DYN | $10^2$–$10^4$ | Leigh et al. [30] |
| EKL | 0.6–15 | Petrovich and Antonini [394] |
| EKL | <100 | VanLandingham et al. [522] |
| EKL | 0.1–48 | Antonini and Perets [379] |
| DYN | 0.21 | O'Leary et al. [264] |
| AGN | | |
| GAS+TRP | 27–37 | Li [523] |
| DYN+GAS | 6–19 | Ford and McKernan [524] |
| MIG+TRP | 0.66–120 | Secunda et al. [525] |
| GAS | 0.002–18 | Gröbner et al. [526] |
| DYN+GAS | 0.02–60 | Tagawa et al. [33] |
| DYN+TRP | 4 | Yang et al. [31] |
| MIG | $10^{-3}$–$10^4$ | McKernan et al. [514] |
| TRP | $10^2$–$10^4$ | Leigh et al. [30] |
| DYN+GAS | 3 | Stone et al. [442] |

**Table 2.** Same as in Table 1, but for NS–BH binaries. Note that the estimate from McKernan et al. [521] was obtained under the extreme assumption that all BBH mergers originate in AGNs.

| NS–BH Merger Rates | | |
|---|---|---|
| **Process** | **$\Gamma$ (yr$^{-1}$Gpc$^{-3}$)** | **Ref.** |
| GWTC-3 | | |
| - | **7.8–140** | **The LIGO Scientific Collaboration et al. [141]** |
| Quiescent SMBH | | |
| DYN | $3 \times 10^{-5}$–0.006 | Arca Sedda [136] |
| EKL+SEV | 0.025–0.3 | Wang et al. [359] |
| EKL | 0.06–1 | Fragione et al. [401] |
| DYN | $9 \times 10^{-5}$–0.015 | Arca Sedda [130] |
| DYN | 0.001–0.06 | Hoang et al. [311] |
| AGN | | |
| MIG+TRP | 10–300 | McKernan et al. [521] |
| DYN+GAS | 1.1–6.3 | Yang et al. [134] |

It is interesting to note that the estimated rates for BBH mergers in galactic nuclei are comparable to those predicted for other channels (for a complete review on merger rates from different channels, see [527]). Therefore, untangling the signatures of different formation channels in ground-based observations might be a quite hard task.

Future detectors might help in unravelling how different channels contribute to the overall population of BBH mergers. An equal-mass binary with total mass $m_{\rm bin}$ emitting GWs at frequency f will merge in a timescale:

$$\tau \sim 5\,{\rm yr} \left(\frac{f}{10\,{\rm mHz}}\right)^{-8/3} \left(\frac{m_{\rm bin}}{68\,{\rm M}_\odot}\right)^{-8/3};\qquad(45)$$

thus, combining LISA and ground-based observations has the potential to cover the full BBH frequency spectrum. Sources in the mass range between GW150914, the first GW sources ever detected [65], and GW190521 [76], emitting at 10 mHz, could be observed in the last 5–1 yr prior to merger, enabling more precise measurement of the localisation, eccentricity, spin amplitude and alignment, and binary mass [445,448]. More likely, ground-based observations can be *reverse-engineered* by seeking the observed binaries in data previously acquired with LISA. Unfortunately, recent models suggest that LISA will observe only a few tens to a hundred BBHs, with only $\lesssim 10$ potentially observable as multi-band sources, depending on LISA's mission lifetime and the adopted SNR threshold [445,448,489,528,529].

Deci-Hz observatories [452–455], filling the gap between space-borne and ground-based detectors, could permit us to pierce deeper into the cosmos and reconstruct BBH mergers' evolutionary history. A DECIGO-like detector [453,454] has the potential to observe merging BBHs up to a redshift $z \sim 10^3$ [455], well beyond the formation time of the first stars. A more modest detector design, although still very ambitious, could still provide the observation of stellar BBH mergers up to redshift $z = 3 - 100$ [455].

*8.3. Imprint of Galactic Nuclei on the Gravitational Wave Emission from Merging Compact Objects*

An SMBH in the vicinity of a merging BBH can leave several imprints, some of which will be potentially measurable with future space-borne and third-generation detectors.

8.3.1. Eccentricity Variation Encoded in the Gravitational Wave Signal of Merging Compact Objects

LISA has the potential capability to track signatures of the EKL mechanism in our own Galactic Centre [490,530,531]—and nearby galactic centres [532]—and probe the very existence of compact BBHs orbiting around an SMBH. As we described in the previous sections, a triple system composed of an inner binary whose centre of mass orbits around a distant perturber can be subjected to secular oscillations of the orbital inclination and eccentricity of the inner orbit.

A compact binary with mass $m_{\rm bin}$, semimajor axis $a$, and eccentricity $e$ emits GWs characterised by a broad frequency spectrum peaking at [264]

$$f_{\rm peak} = 7.6 \times 10^{-6}\,{\rm Hz}\,\frac{(1+e)^{1/2}}{(1-e)^{3/2}} \left(\frac{a}{0.1{\rm AU}}\right)^{-3/2} \left(\frac{M}{50\,{\rm M}_\odot}\right)^{1/2} \sim 0.01\,{\rm Hz}.\qquad(46)$$

From the equation above, it is apparent that a circular binary with mass $m_{\rm bin} = (25 + 25)\,{\rm M}_\odot$ and semimajor axis $a = 0.1$ AU—which implies a merging time of $t_{\rm GW} = 1\,{\rm Gyr}$ [388]—would be invisible to LISA until its semimajor axis reduces by a factor of $O(10)$. However, the EKL mechanism at play can trigger eccentricity oscillations, which can bring the GW peak frequency into the LISA band well before the binary shrinks, provided that the eccentricity peak reaches a sufficiently large value, $e_{\rm peak} > 0.95$, in the aforementioned example. Additionally, EKL will shorten the binary lifetime by a factor of $10^4$ [379,491].

Such a binary moving about an SMBH like SgrA* will undergo a periodic shift of the frequency peak from $10^{-5}$ Hz up to 0.1 Hz [490], thus fully covering the LISA sensitivity band. The period of such a shift—$\sim 1$ month—can be much shorter than the LISA mission's lifetime, thus making possible the detection of such a clear signature of a BBH–SMBH triple (see also [530,531]). EKL signatures could be observed also in nearby galactic nuclei, at a distance of 1–10 Mpc, depending on the binary and SMBH masses and the orbital properties [490,531,532].

The same process might help in spotting binaries containing an IMBH around an SMBH [532,533].

Detection (or not) of even one such signatures would provide crucial insights into the formation probability of BBHs around an SMBH and the possible contribution of galactic nucleus BBHs to the overall population of BBH mergers.

Several deci-Hz observatories, such as DECIGO or the Big Bang Observatory, could enable an even clearer detection of such signatures [531]. Moreover, eventual synergies among different detectors operating at the same time could enable a simultaneous tracking of the EKL mechanism at play, helping to remove any possible degeneracy in the parameter space [531].

### 8.3.2. Supermassive Black Hole Acceleration Encoded in the Gravitational Wave Signal of Merging Compact Objects

In general, merging BBHs emit GWs at a frequency that is redshifted by a factor f $(1 + z_C)^{-1}$, with $z_C$ being the cosmological redshift. However, the signal emitted by the BBH while revolving around the SMBH suffers a time-dependent variation caused by its motion with respect to the observer.

The variation in the distance between the emitter and the observer causes a Doppler shift in the arrival time of GW pulses; the gravitational time dilation causes a relativistic Doppler shift and a gravitational redshift, whilst the SMBH gravitational field causes a delay in the time of arrival of the GW signal. The aforementioned three effects, called Roemer, Einstein, and Shapiro delay [534], cause a shift in the measured GW signal, which might be measurable in both low- [499,535–540] and high-frequency detectors [165,541,542].

For example, LVC detectors could measure such shift in BBH binaries with masses $m_{\mathrm{bin}} \gtrsim 20\,\mathrm{M}_\odot$ orbiting at $r \sim 1$ AU from an SMBH with mass $M_{\mathrm{SMBH}} \sim 10^5$–$10^6\,\mathrm{M}_\odot$ [541].

The Doppler boost effect has potentially crucial implications for the interpretation of detected signals. In fact, aside from the cosmological redshift, the signal emitted by a BBH revolving around an SMBH can suffer a Doppler shift:

$$(1 + z_D) \sim (1 - v^2/c^2)^{-1/2}(1 + v/c),$$

and a gravitational redshift [165,537,538,541–543]. Defining the SMBH Schwarzschild radius ($R_S$) and the semimajor axis of the BBH orbit around the SMBH ($r$) enables us to calculate the Doppler shift at the pericentre through the relation $(v/c)^2 = (R_S/2r)(1 + e)/(1 - e)$ and the gravitational redshift as

$$(1 + z_G) \sim (1 - R_S/r)^{-1/2},$$

under the simplistic assumption that the GW emitter and receiver are on the same side with respect to the SMBH [542]. Thus, the total shift induced by the SMBH can be written as $(1 + z) = (1 + z_C)(1 + z_D)(1 + z_G)$ [165,542]. Since the measured BBH chirp mass scales with $\mathcal{M}_{\mathrm{obs}} \propto f^{-11/5}\dot{f}^{3/5}$ and its luminosity distance $D_{\mathrm{L,obs}} \propto f^{2/3}$, the measured and intrinsic quantities are linked by the total redshift as

$$
\begin{align}
\mathcal{M}_{\mathrm{obs}} &= \mathcal{M}(1 + z_C)(1 + z_D)(1 + z_G) \tag{47}\\
D_{\mathrm{L,obs}} &= D_C(1 + z_C)(1 + z_D)(1 + z_G), \tag{48}
\end{align}
$$

where $D_C$ is the BBH comoving distance. Depending on the values of the additional corrections, the interpretation of observed sources might thus be biased toward larger values [542]. Note that this corrective factor depends only on the cosmological location of the merger, the SMBH mass, and the BBH–SMBH orbital semimajor axis and eccentricity. Figure 16 shows how the correction factor varies with the SMBH mass and the distance to the galactic centre assuming that the merger occurs at redshift $z_C = 0$ and the BBH–SMBH orbital eccentricity $e = 0 - 0.5 - 0.9$. Note that the spikes occur when $R_S \sim r$ or $R_S \sim 2r(1 - e)/(1 + e)$. The picture makes evident that both the observed chirp

mass and luminosity distance might overestimate the intrinsic properties of the merger by a factor of up to 2–3 [542] and on average by around 10–30% [165].

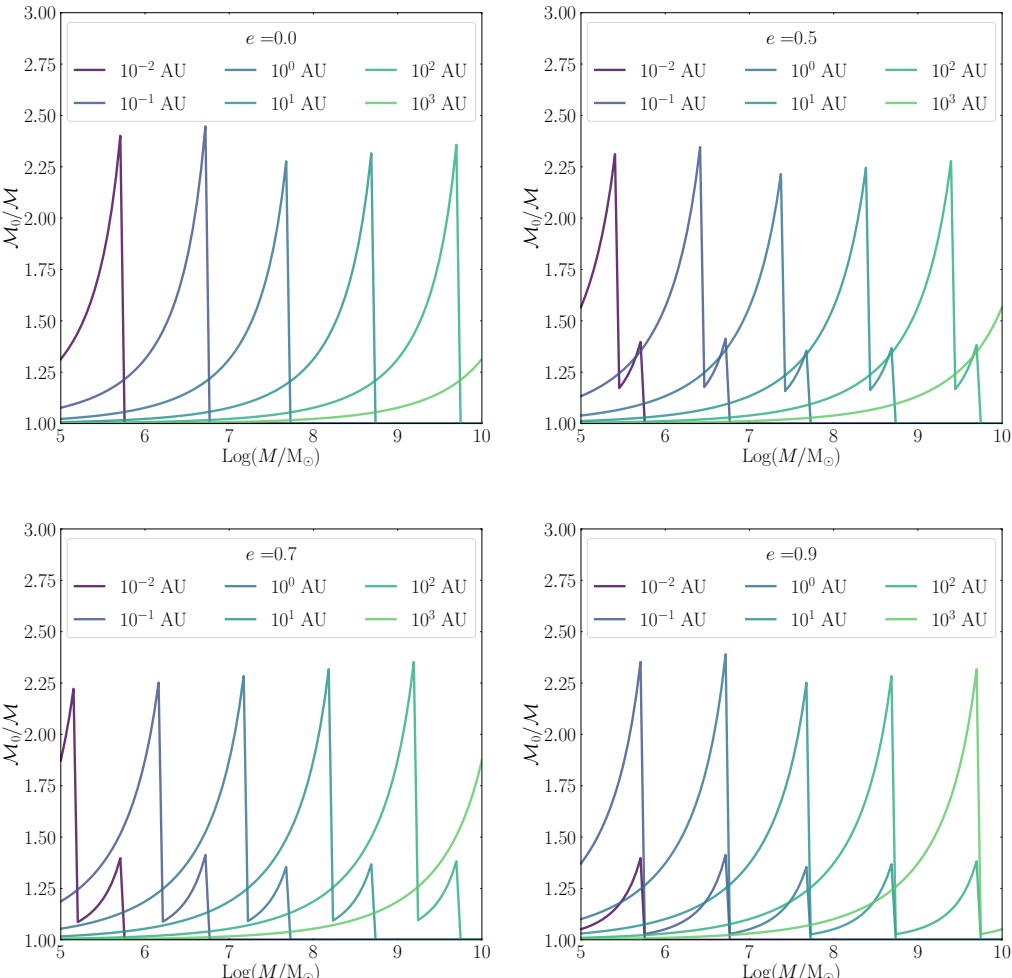

**Figure 16.** Total chirp mass correction factor as a function of the SMBH mass and for different values of the BBH–SMBH separation, assuming that theBBHs undergo a merger at redshift $z = 0$ and move around the SMBH on an orbit with an eccentricity $e$= 0–0.5–0.7–0.9.

It is worth noting that a BBH with mass $m_{bin} = 50$ M$_\odot$ at a distance $r = 1$ mpc from an SMBH with mass $M_{SMBH} = 10^6(10^8)$ M$_\odot$ revolves around the SMBH in a time of $P_2 \simeq 3(0.3)$ yr; thus, a detector with a mission lifetime ($\tau_{life}$) longer than the BBH merging time and the orbital period could measure the Doppler modulation in the source signal [535–537,540] and related effects [499,539,543,544].

While revolving around the SMBH, the source will appear to change its redshift by a factor of $\delta z \sim a_{COM}\tau_{life}/c$ [535,536]. The resulting progressive drift of the GW signal could be detected by LISA in principle, and since it is maximised in galactic nuclei, its detection could represent a strong indicator of the merger environment [535]. The effect of Doppler boost could be measurable up to a redshift $z < 0.05$, thus permitting probing the BBH–SMBH interplay in the local Universe [537], but even in the case of no detection, the acceleration drift could be sufficient to bias the parameter estimation of future detected sources [536]. A sufficiently long monitoring of the source could help measuring the SMBH mass [545] and also lead to the detection of lensing signatures caused by the SMBH [539,543,546]. The probability to observe such an effect depends on the properties of BBHs forming around an SMBH and could be O(1–3%) with both LISA and TianGO [539,543], potentially enabling measuring the SMBH's mass at a level

of 0.01–1% [539,540]. Similar to electromagnetic wavelets, GWs from BBHs around an SMBH could be affected by aberration, an effect that can cause a shift in the GW signal as significant as the "standard" Doppler shift and potentially detectable by a LISA-like mission, provided that the BBH is orbiting within a few $10^3$ Schwarzschild radii [544]. The detection of a few hundred BBHs should thus help us start to unveil how different channels contribute to the formation of merging BBHs [543].

Furthermore, the GW signal emitted by the merging COB can be scattered by the central SMBH. This can produce a secondary signal, a sort of "GW echo", which will have a similar time–frequency evolution as the main signal, but is delayed in time. For sources moving within $10$–$10^4$ Schwarzschild radii from the SMBH, around $10(90)\%$ of the detectable echo arrives within $\sim 1(100)\text{s}(M_{\text{SMBH}}/10^6\,\text{M}_\odot)$ after the primary signal [547].

In the case of Kerr SMBHs, there are three further effects that might leave an imprint on the BBH-emitted signal, namely (i) the precession of the BBH–SMBH angular momentum around the SMBH spin owing to spin–orbit coupling associated with the 1.5 PN effect [384,497,500], (ii) the precession of the BBH angular momentum around the BBH–SMBH angular momentum, a geodesic de-Sitter-like precession induced by GR [548,549], and (iii) precession of the BBH angular momentum around the SMBH spin owing to the Lens–Thirring precession. The coupling of these effects can lead to a modulation in the GW signal of inspiralling BBHs, possibly being observable with space-borne detectors such as LISA if the precession period is shorter than the observation time [499,500].

Finally, if the BBH is travelling in an AGN disc, the gaseous medium can leave an imprint on the binary waveform, which translates into an overestimate of the binary chirp mass by a factor of $\left(1 + \tau_{\text{gw}}/\tau_{\text{gas}}\right)^{3/5}$ and of the binary distance by a factor of $\left(1 + \tau_{\text{gw}}/\tau_{\text{gas}}\right)$, where $\tau_{\text{gw}}$ and $\tau_{\text{gas}}$ represent the GW and the hydrodynamical timescale, respectively [550].

## 9. Summary

Galactic nuclei likely represent the most-intricate environments in the Universe, where the dynamics is regulated by a complex interplay among stellar interactions powered by the high densities of NCs, secular effects driven by central SMBHs, and gas-driven effects in AGNs.

Discoveries such as the GWs emitted by merging COs, the high-energy emission from the Galactic Centre possibly triggered by COBs, the stellar motion in the immediate vicinity of SgrA*, and the Galactic SMBH recently revived the interest in how COs form, pair-up, and eventually merge in quiescent and active galactic nuclei harbouring an NC, an SMBH.

In this review, we provided an overview of the plethora of mechanisms that can contribute to the formation, and possibly coalescence, of COBs in both galactic nuclei with a quiescent SMBH and AGNs. These mechanisms rely on different theoretical frameworks devised to represent different aspects—e.g., scatterings, EKL, gas torques—of the same environment.

We highlighted the main imprints that one mechanism or another could leave in the population of COBs it produces and how they could be used to untangle the origin of some GW sources detectable with present-day or future GW observatories.

The main peculiarities of the most-recent theoretical frameworks can be summarised as follows:

- The dynamics plays a crucial role in determining COB formation in galactic nuclei. Three-body scatterings, involving three initially unbound objects, are likely dominant in galaxies with a large NC-to-SMBH mass ratio, but become extremely inefficient close to the SMBH. Conversely, single–single interactions that form bound pairs via GW bremsstrahlung—or GW captures—are more efficient in the SMBH's immediate vicinity and in the nuclei with the most-massive SMBHs. However, GW captures produce short-lived binaries that merge within days or hours from their formation and have a large chance of being highly eccentric when sweeping through high-frequency detectors.

- Galaxies dominated by a quiescent SMBH can efficiently replenish their population of COBs—particularly BHs—via the accretion of massive star clusters that undergo inward migration owing to dynamical friction.
- A substantial population of primordial binaries can also play a crucial role in determining the properties of COBs in galactic nuclei, although most of them are likely destroyed by the SMBH's tidal field.
- Once binaries start forming in galactic nuclei, their further evolution is regulated mostly by binary–single interactions, which generally promote the formation of tighter and more massive binaries, but, depending on the binary properties, can lead to their evaporation well before GW emission starts dominating the binary evolution.
- Owing to dynamical friction, or mass segregation, and dynamical interactions, COBs are expected to move through regions of the nucleus with different velocity dispersions and densities. The variation of the environment structure can dramatically affect the COB's fate: an initially hard binary moving inward can appear soft closer to the SMBH and eventually be disrupted by interactions with other stars and COs.
- Around 20–70% of COBs formed in galactic nuclei are expected to suffer the effect of the SMBH's gravitational field, which can cause periodic oscillations of their eccentricity called eccentric Kozai–Lidov resonances. This mechanism can significantly shorten the COB's lifetime, possibly affecting the delay time of merging COs. The development of EKL oscillations strongly depends on the binary properties (e.g., general relativistic precession can suppress EKL), the distance to the SMBH, and the eccentricity of the COB's orbit about the SMBH.
- In AGNs, the formation of COBs is favoured by both gaseous torques and dynamical scatterings, whose efficiency is boosted by the nearly planar configuration. The possible existence of migration traps, where inward and outward torques cancel out, makes AGNs potential factories of multiple-generation COs mergers and IMBHs.
- Mergers occurring in galactic nuclei feature some peculiar traits: a significant fraction of mergers with one component in the upper mass-gap, a non-negligible fraction of multiple-generation mergers that can affect the high-end of the BH mass distribution, and fairly misaligned spins; although, in AGNs, a noticeable fraction of high-generation mergers might have mildly aligned spins and a quite significant probability to preserve an eccentricity $e > 0.1$ whilst sweeping through the frequency bands of both low- and high-frequency detectors.
- The merger rate inferred for present-day GW detectors for BBH and NS–BH binary mergers in galactic nuclei is poorly constrained owing to the many model uncertainties. For BBH mergers, models for quiescent SMBHs and AGNs predict similar estimates, which generally fall in the range $\mathcal{R}_{\mathrm{BBH}} = 10^{-3} - 10^2 \mathrm{yr}^{-1} \mathrm{Gpc}^{-3}$. For NS–BH mergers, instead, there are clear differences between the prediction for quiescent, $\mathcal{R}_{\mathrm{BBH}} = 10^{-5} - 1 \mathrm{yr}^{-1} \mathrm{Gpc}^{-3}$, and active nuclei models, $\mathcal{R}_{\mathrm{BBH}} = 1 - 10^3 \mathrm{yr}^{-1} \mathrm{Gpc}^{-3}$, partly owing to the relatively poor literature and the huge uncertainties.
- The presence of an SMBH in the vicinity of a merging COB can leave some imprints on the emitted GW signal that could, in principle, be detected with future detectors, among others a shift in the peak frequency for mergers occurring in the Milky Way centre, a variation in the measured redshift induced by the rapid motion of the binary around the SMBH, and the development of a GW echo produced by the scattering of the emitted GWs onto the SMBH.

Despite the current literature being very rich and the amount of new work performed in the field constantly increasing, there are still a number of important elements that are missing in current models and that deserve further development. Among others, we identify in the following some elements that may be key to place more stringent constraints on the model predictions:

- **Initial conditions**: Probably the most-important unknown that mostly affects all the models is the scarce knowledge of how stars form and pair in the extreme environment of a galactic nucleus. The initial binary fraction, the initial distribution of periods and

masses, and the metallicity spread in the galactic nucleus are all factors that crucially determine the COB's properties: semimajor axis, eccentricity, and component masses.

- **Interplay of mechanisms**: As we have seen throughout the review, COB formation is likely regulated by many mechanisms likely operating simultaneously. However, most theoretical models focus on one specific aspect at a time. Fully self-consistent *N*-body simulations capable of taking into account the stellar evolution of single and binary stars, the SMBH tidal field, and potentially, the effect of an AGN disc exist, but their resolution is still too low and their computational cost too large to permit a one-to-one representation of a Milky Way-like nucleus. Simpler models relying on semi-analytic assumptions or few-body (scattering) simulations represent valid alternatives, although they sometimes neglect potentially crucial elements, such as the importance of flybys on the evolution of COBs undergoing EKL oscillations, the development of EKL resonances in binaries formed in AGNs, and the role of star formation in the actual population of COBs around an SMBH.

- **Observations**: From an observational perspective, the observation of young massive binaries in galactic nuclei, a larger number of GW detections, a more precise localisation of GW sources, and the future detection of inspiralling binaries in the Milky Way centre can definitely help us improve our knowledge of the processes regulating COB formation in galactic nuclei.

**Author Contributions:** M.A.S., S.N. and B.K. have equally contributed to the conceptualisation, data collection, writing, and editing of this review. All authors have read and agreed to the published version of the manuscript.

**Funding:** M.A.S. acknowledges funding from the European Union's Horizon 2020 Research and Innovation Programme under the Marie Skłodowska-Curie Grant Agreement No. 101025436 (Project GRACE-BH, PI: Manuel Arca Sedda). S.N. acknowledges the partial support from NASA ATP 80NSSC20K0505 and the NSF through Grant No. 2206428, and thanks Howard and Astrid Preston for their generous support. This work was supported by the Science and Technology Facilities Council Grant Number ST/W000903/1 (to B.K.).

**Institutional Review Board Statement:** Not applicable.

**Informed Consent Statement:** Not applicable.

**Data Availability Statement:** Not applicable

**Conflicts of Interest:** The authors declare no conflict of interest.

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
