# Peer review of "Quiescent and Active Galactic Nuclei as Factories of Merging Compact Objects in the Era of Gravitational Wave Astronomy"

_universe, doi:10.3390/universe9030138_

Round 1

Reviewer 1 Report

The review paper entitled "Quiescent and active galactic nuclei as factories of merging compact objects in the era of gravitational-wave astronomy" makes

an encyclopedic and timely effort to gather comprehensively the literature

on the subject, which is going under increasing investigation due to the recent

and expected future detections of gravitational waves from compact

binary coalescences.

The paper embraces a lot of physical effects and have exhaustive references,

but it is hard to read, or better, to digest.

While the outline spelled out in the introduction (and in the summary) is clear, each section appears

not sufficiently linked to the previous/next, reporting a plethora of scattered

results without much explanations.

E.g. around L42 is said that Milky Way galactic center is an "unlikely nursery

for IMBH" without giving any explanation or hints.

The same is done for tens of other formulae in the paper, for which references,

but no explanation, is given.

I understand that including a full derivation for each formula would turn this

review paper into a huge book, but at least some qualitative, explanatory

remark could be added.

It would also help the reader to add some small transition texts from one

section to the other, explaining how the physical effect being successively

considered

(evaporation, effect of nuclear cluster, dynamical friction, mass segregation,

just to name the first ones) fit into the big picture outlined in fig. 1, and

are related to one another.

Moreover, a more detailed scheme than Fig. 1 and/or a table summarizing the

path from star clustering to compact binary coalescence would be helpful.

Minor issues:

* There are few acronyms for which I could not find definitions (ZAMS, OB, WR).

  Given the large number of acronyms adopted, a table summarizing all of them

  would be helpful.

  Also I do not think it is necessary to define acronyms in the abstract

  and to repeat their definitions in the text.

* There are a few typos and a few un-latexed references

* Sec. 8.2 is empty (it has probably been replaced by "Expected merger rate and prediction")

Author Response

Dear Editor, 

we are grateful to the referee for their insightful comments and suggestions. Following them, we decided to reshape and re-organise the review in such a way as to make it easier to digest for the reader. 

We also added text to explain more some of the formulae and their implications. 

We addressed all minor issues raised by the referee, highlighting all the new and reshaped text in boldface for the sake of clarity.

Best regards,

Manuel, on behalf of the authors

Reviewer 2 Report

The submitted text "Quiescent and active galactic nuclei as factories of merging compact objects in the era of gravitational-wave astronomy" is trying to estimate number of gravitational wave events generated by compact object mergers. The article is well written and deals with currently very relevant subject, so I can recommend it for Universe journal.

There are only some missing citations on lines: 244,914 and 915.

Author Response

Dear Editor, 

we thank the referee for the nice words.

We have fixed all missing references, thanks for noticing it.

Best wishes,

Manuel, on behalf of the authors